



**Three-dimensional transient flow to a partially penetrated well with variable discharge**
**in a general three-layer aquifer system**
Qinggao Feng[1]*, Xiaola Feng[1], Hongbin Zhan[2]*
[1]Faculty of Engineering, China University of Geosciences, Wuhan, Hubei, 200092, P. R.
China
[2]Department of Geology and Geophysics, Texas A&M University, College Station, TX
77843-3115, USA.
*Corresponding authors
Emails: fengqg@cug.edu.cn (Q. Feng), zhan@geos.tamu.edu (H. Zhan)



**ABSTRACT**
A general analytical model for three-dimensional flow in a three-layered aquifer system with
a partial penetration well having a variable discharge of pumping is developed by taking
account of the interface flow on the adjacent layers. This general three-layer system includes
the conventional aquitard-aquifer-aquitard system as a subset and does not require that the
permeability contrasts of different layers must be greater than a few orders of magnitude, and
does not ignore any flow components (either vertical or horizontal) in any particular layer.
The pumping well of infinitesimal radius is screened at any portion of the middle layer. Three
widely used top and bottom boundary conditions are considered that can be specified as a
constant-head boundary (Case1) or a no-flux boundary (Case 2), and a constant-head
boundary at the top in combination with a no-flux boundary at the bottom (Case 3). Laplace
domain solutions for dimensionless drawdown are obtained by the use of Hankel
transformation, and associated time-domain solutions are evaluated numerically. The newly
obtained solutions include some available solutions for two- or single-layer aquifer systems
as subsets. The drawdowns for individual layers caused by a well with an exponentially
decreased discharge are explored as an example of illustration. The results indicate that the
pumped layer drawdown close to the partially penetrated well is mainly influenced by the
variable pumping rate. The late-time drawdowns for all layers are remarkably affected by the
chosen types of top and bottom boundary conditions, and the drawdown for Case 3 is greater
than that for Case 1 and smaller than that for Case 2. Additionally, the effect of the pumped
layer anisotropy on drawdowns in the three-layer system is significant, and the anisotropy of
the unpumped layers significantly affects the drawdown in the whole aquifer system without



large contrast of hydraulic conductivity between the unpumped layers and the pumped layer.
The drawdowns in all three layers are greatly affected by the location and length of well
screen, and a larger drawdown can be seen at the position that is closer to the middle point of
the screen of the partially penetrating pumping well.
*Keywords*: Three-layer system; Well partial penetration; Variable discharge; Top and bottom
boundary; Semi-analytical solution.



## 1. **Introduction**

Most groundwater flow model concerning a pumping and/or injection well will have the
pumping and observation wells in the same aquifer (Yeh and Chang, 2013; Houben, 2015).
For a multi-aquifer system, the pumping and observation wells may be in the same aquifer or
in different aquifers. As different aquifers in a multi-aquifer system are hydraulically
connected, pumping in a specific aquifer will inevitably induce hydraulic responses over the
entire multi-aquifer system, and the observation well in an unpumped aquifer will also record
the drawdown information associated with pumping in the pumped aquifer. Therefore, the
questions we need to answer are: How to interpret the drawdown information collected at an
unpumped aquifer from the pumped aquifer? And furthermore, is that feasible to conduct
aquifer characterization and to obtain the aquifer hydraulic parameters when the drawdown
information is collected at an unpumped aquifer from the pumped aquifer? To answer these
questions, one must first develop a robust groundwater flow model in a fully coupled
multi-aquifer system. Unfortunately, the present models on this subject are severely limited to
some demanding and often time unrealistic restrictions.
The present groundwater flow models related to multi-layer aquifer systems are usually
established by solving the coupled partial differential equation group of groundwater flow
explicitly or with a matrix solver (Bakker, 2013; Chen and Morohunfola, 1993; Cihan et al.,
2011; Hantush, 1967; Hunt, 2005; Meonch, 1985; Neuman and Witherspoon, 1969). In those
models, some strong assumptions are often invoked to simplify the system. For instance, it is
commonly assumed that the permeability contrasts among two adjacent aquifers are more
than a few orders of magnitude, thus flow in the much less permeable layer is assumed to be
perpendicular to the layering while the flow in the much greater permeability layer is
assumed to be parallel to the layering (Hantush, 1967; Neuman and Witherspoon, 1969).
Such a simplification may be acceptable for investigating an aquifer-aquitard system as the
aquitard/aquifer permeability contrasts can be indeed as large as a few orders of magnitude
(Hantush, 1964; Lin et al., 2019; Neuman, 1968; Yeh and Chang, 2013). But this assumption
is baseless for a general multi-aquifer system in which the permeability contrasts among
different layers are much modest. Another commonly used assumption in present models is
that mass exchange between two adjacent aquifers can be treated as a volumetric sink/source
incorporated into the governing equations of flow in each individual layer (the so-called
Hantush-Jacob assumption) (Hantush and Jacob, 1955). This assumption is also problematic
in the sense that it does not honor the fact that mass exchange between two adjacent layers
always occurs at the interfaces of those adjacent layers rather than as a volumetric sink/source
inside those layers, a treatment that can generate considerable errors, as documented in
numerous investigations (e.g. Hantush, 1967; Feng and Zhan, 2015; Feng et al., 2019, 2020;
Zhan and Bian, 2006). A third simplification in present models is to assume a constant
pumping rate (Hantush, 1964; Yeh and Chang, 2013). The constant pumping rate is desirable
but is quite difficult to maintain in actual pumping scenarios which almost always involve
variable pumping rates because of many reasons such as the temporary loss of power,
increased drawdown in the pumping well with time (which makes it more difficult to lift
water from the pumping well) and other constrains in conducting pumping tests in the field
(Chen et al., 2020; Hantush, 1964; Mishra et al., 2013; Sen and Altunkaynak, 2004; Singh,
2009; Wen et al., 2017).



In theory, numerical modeling can avoid many restrictions mentioned above to
investigate a multi-aquifer system, but it has some issues that are sometimes not easy to
resolve. For instance, it is not straightforward to use a numerical model for aquifer
characterization to obtain the aquifer parameters, particularly when dealing with a
multi-aquifer system involving many hydraulic parameters for multiple aquifers. When the
numerical model has to be used for such a purpose, it often involves either trial-and-error or
automatic optimization procedures to minimize the model-generated drawdown with the
observed drawdown (Mohanty et al., 2013; Jeong and Park; 2019; Rajaee, et al., 2019). This
process can sometimes lead to non-uniqueness of inverted aquifer parameters (Rahman et al.,
2020). Another issue associated with numerical model is that without a benchmark analytical
solution, it is unknown how much numerical errors have been involved in the numerical
model. For a multi-aquifer system, the numerical errors can be considerable near the
interfaces of different aquifers where the aquifer parameters change suddenly (Neuman, 1968;
Louwyck et al., 2012). If one recalls that any numerical approaches (no matter they are
finite-difference, finite-element, boundary-element, or others) essentially involve some sorts
of smoothing or average schemes to approximate the mass conservation law in a discrete
sense, then it is not surprise to know that numerical errors are prone to be large near sharp
interfaces (Cihan et al., 2011; Neuman, 1968; Li and Neuman, 2007; Loudyi et al., 2007). Of
course, one can use gradually finer meshes when approaching the interfaces of different
aquifers to minimize the numerical errors, but such a procedure can sometimes increase the
computational cost rapidly, particularly when dealing with three-dimensional (3D) flow in a
multi-aquifer system (Feng et al., 2020; Rajaee, et al., 2019; Rahman et al., 2020; Rühaak et





al., 2008). Overall, establishing a sufficiently accurate numerical model for groundwater flow
in a multi-aquifer system is feasible, but often time requires considerable preparations and
computational cost.
Based on above considerations, we are going to establish a robust and generic 3D
groundwater flow in a three-aquifer system in this investigation. The generality of this work
is reflected on the following aspects. Firstly, it does not put any constrains on the
permeability contrasts among different aquifers involved. Such a generality will make this
work much more appealing to deal with a vast number of cases in actual aquifer setting. It
also encompasses previous aquifer-aquitard two-layer system and aquitard-aquifer-aquitard
three-layer systems as subsets. It can even be applied for an extreme two-layer or three-layer
system such as a fracture-rock two-layer system or a rock-fracture-rock three-layer system
when flow can occur in both fractures and rock matrix. Furthermore, for the
rock-fracture-rock three-layer system, the rocks adjacent to the fracture can be either identical
with the same hydraulic properties or have different lithology and hydraulic properties. The
two-aquifer system investigated by Feng et al. (2019) is also a subset of this study. Secondly,
this study honors the mass exchange among different aquifers as an interface flow
phenomenon, not as a volumetric sink/source term, as in the Hantush-Jacob assumption.
Thirdly, the pumping rate can be any given function of time instead of being a constant. This
is a distinctive difference from the three-aquifer study of Feng et al. (2020) involving
constant pumping rate. Fourthly, three widely used top and bottom boundary conditions are
considered that can be specified as a constant-head boundary (Case1) or a no-flux boundary
(Case 2), and a constant-head boundary at the top in combination with a no-flux boundary at



the bottom (Case 3). This is also in contrast with Feng et al. (2019, 2020) which cannot
investigate the combined effects of the top and bottom boundaries simultaneously. In the
following sections, semi-analytical drawdown solutions in nondimensional forms in a genetic
three-layer system are obtained by performing Laplace-Hankel transform and eventually the
real time solutions are calculated by the method of numerical inversion. Finally, as an
example of illustration, the characteristics of drawdown are thoroughly investigated due to a
partially penetrated well pumped at an exponentially decreased discharge function. The
results are discussed extensively and their applications are elaborated as well.
**2. Methodology**
**2.1 Mathematical model**

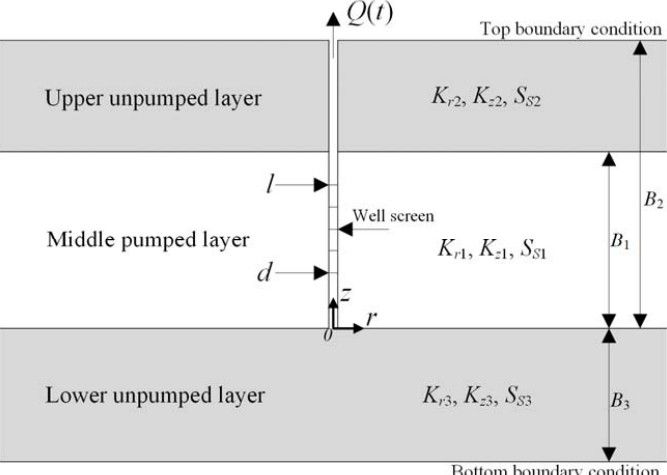


Fig.1 Schematic diagram of a three-layer aquifer system with a partial penetration well

Fig. 1 displays an infinitesimal-radius well with a variable discharge $Q(t)$ in a general

three-layer aquifer system of unbound lateral extension. The pumping well is partially
penetrated in the middle layer of the system with a screen length from $d$ to $l$ shown in this



figure. Each layer of constant thickness is homogeneous and anisotropic. Three-dimensional
flow is included in all layers. The interface flow at the two neighboring layers is linked with
head and flux continuity conditions. It is noted that three different cases presented by
Hantush (1960) are concluded, specifically, the boundaries at the top and bottom are
simultaneously constant-head boundaries (Case 1), no-flux boundaries (Case 2), or a
combination of a constant-head top boundary and no-flux bottom boundary (Case 3). The
cylindrical coordinate origin is at the intersection of the well axis and the bottom of the
middle-pumped layer.

According to the conceptual model above, the equations that govern the transient

drawdown distribution for flow to a pumping well can be given by:
$\dfrac{K_{ri}}{r}\dfrac{\partial}{\partial r}\left(r\dfrac{\partial s_i(r,z,t)}{\partial r}\right)+K_{zi}\dfrac{\partial^2 s_i(r,z,t)}{\partial z^2}=S_{si}\dfrac{\partial s_i(r,z,t)}{\partial t}$     (1)
where $s(r,z,t)$ denotes drawdown at space coordinate (radial distance $r$ [L], vertical distance
$z$ [L]) and time coordinate (pumping time $t$ [L]); $K_r$ and $K_z$ indicate, respectively, the
hydraulic conductivities in the radial and vertical direction [L/T]; $S_s$ refers to specific storage
[1/L], and    $i = 1, 2, 3$ designate, respectively, the middle-pumped layer, upper layer and
lower layer.

The initial conditions of the aquifer system can be written as:

$s_i(r,z,0)=0$    (2)

The boundary of the aquifer system at infinity yields:

$s_i(\infty,z,t)=0$    (3)

The pumping well of infinitesimal diameter is partially penetrated in the middle layer,

the wellbore boundary condition is subject to (Hantush, 1964, Liang et al, 2018):



$$\lim_{r \to 0} r \frac{\partial s_1}{\partial r} = \begin{cases} 0 & l < z \leq B_1 \\ -\dfrac{Q(t)}{2\pi K_{r,1}(l-d)} & d \leq z \leq l \\ 0 & 0 \leq z < d \end{cases} \quad (4)$$
in which $Q(t)$ represents the well discharge of pumping [L$^3$T$^{-1}$], $B_1$ refers to the thickness of
the middle-pumped aquifer [L]. It is notable that an assumption of the well discharge
uniformly distributed along the screened section of the partially penetrating well is used
herein. This, of course, is a simplification for the sake of mathematical modeling. Fortunately,
this simplification is proven to be sufficiently accurate for regions that are not extremely
close to the pumping well (within a few well radii) (Chang and Yeh, 2013).

As an example of illustration, the pumping rate used in this study varies exponentially

with the pumping time in the form (Hantush, 1964b, 1966; Wen et al., 2017):
$Q(t) = Q + (Q_1 - Q)e^{-\alpha t}$    (5)
which is based on lots of field data and available works (Chen et al., 2020; Feng et al., 2019;
Sen and Altunkaynak, 2004). The symbol $Q$ and $Q_1$ represent the final (constant) and initial
well discharge, respectively [L$^3$T$^{-1}$], and $\alpha$ designates decay constant obtained from the
measured data of pumping [T$^{-1}$].

The inner well-face boundary conditions at the upper and lower unpumped layers yield:

$\lim_{r \to 0} r \dfrac{\partial s_2}{\partial r} = \lim_{r \to 0} r \dfrac{\partial s_3}{\partial r} = 0$    (6)

And the boundary condition at the interface between the middle-pumped aquifer and the

adjacent upper layer ($z = B_1$) requires that:
$s_1(r,z,t) = s_2(r,z,t), \quad z = B_1$    (7)
and





$K_{z,1} \dfrac{\partial s_1(r,z,t)}{\partial z} = K_{z,2} \dfrac{\partial s_2(r,z,t)}{\partial z}, \quad z = B_1$     (8)
The continuity of hydraulic connection between the middle-pumped layer and the lower
unpumped layer ($z = 0$) can be written as:
$s_1(r,z,t) = s_3(r,z,t), \quad z = 0$     (9)
and
$K_{z1} \dfrac{\partial s_1(r,z,t)}{\partial z} = K_{z3} \dfrac{\partial s_3(r,z,t)}{\partial z}, \quad z = 0$     (10)
The top boundary condition at the upper unpumped layer ($z = B_2$) and the bottom
boundary condition at the lower unpumped layer ($z = B_3$) of the aquifer system can be, in the
manner of Hantush (1960) and Moench (1985), expressed in three ways.
For Case 1, the constant-head boundaries at both top and bottom boundaries can be
respectively written as
$s_2(r,z,t) = 0, \quad z = B_2$     (11)
and
$s_3(r,z,t) = 0, \quad z = -B_3$     (12)
For Case 2, the no-flux boundary at both top and bottom boundaries yield
$\dfrac{\partial s_2(r,z,t)}{\partial z} = 0, \quad z = B_2$     (13)
and
$\dfrac{\partial s_3(r,z,t)}{\partial z} = 0, \quad z = -B_3$     (14)
For Case 3, the constant-head boundary at the top and the no-flux boundary at the bottom are
respectively
$s_2(r,z,t) = 0, \quad z = B_2$     (15)
and





$\dfrac{\partial s_3(r,z,t)}{\partial z} = 0, \ z = -B_3$    (16)
It should be remarked that the adopted three different types of top and bottom
boundaries expressed in Eqs. (11)–(16) are commonly encountered in practice. In some cases,
the upper layer is covered with ponded water, the upper and lower layers are, respectively,
overlain and underlain a layer of a highly transmissivity, or the induced drawdown at the
top/bottom boundary is not affected by pumping. Under such conditions, the constant-head
condition can be imposed at the boundary. On the other hand, if there is an impermeable layer
below the lower layer or above the upper layer, the no-flux boundary can be adopted
correspondingly. As for the relevant literature, one may consult Baker (2006), Chen et al.
(2020), Feng et al. (2019, 2020), Feng and Zhan (2015, 2016, 2019), Hantush (1960, 1964),
Hemker and Maas (1987), Hunt (2005), Moehch (1985), Neuman and Witherspoon (1969),
Sepúlveda (2008), Wang et al. (2015) and Wen et al. (2011, 2013).
**2.2 Dimensionless solutions**
**2.2.1 Dimensionless equations**

Table 1 Dimensionless variables and parameters

| | | |
|---|---|---|
| $r_D = r / B_1$ | $\alpha_{ri} = K_{ri} / S_{Si}$ | $\gamma_1 = \kappa_1 \xi_2 / \xi_1$ |
| $l_D = l / B_1$ | $\alpha_{zi} = K_{zi} / S_{Si}$ | $\gamma_2 = \kappa_2 \xi_3 / \xi_2$ |
| $z_D = z / B_1$ | $B_{D2} = B_2 / B_1$ | $s_{Di} = 4\pi K_{r1} B_1 s_i / Q$ |
| $d_D = d / B_1$ | $B_{D3} = B_3 / B_1$ | $\alpha_D = \alpha S_{S1} B_1^2 / K_{r1}$ |
| $t_D = \alpha_{r1} t / B_1^2$ | $\alpha_{Dri} = \alpha_{ri} / \alpha_{r1}$ | $\xi_i^2 = (\alpha_{Dri}\lambda^2 + p) / \alpha_{Dzi}$ |
| $\kappa_2 = K_{z2} / K_{z1}$ | $\alpha_{Dzi} = \alpha_{zi} / \alpha_{r1}$ | $\theta_1 = \xi_2(B_{D2} - 1) + \xi_3 B_{D3}$ |
| $Q_{1D} = Q_1 / Q$ | $\kappa_3 = K_{z3} / K_{z2}$ | $\theta_2 = \xi_2(B_{D2} - 1) - \xi_3 B_{D3}$ |






When dealing with complex hydrodynamic systems such as this study,
nondimensionalization has the advantage of untangling parameter correlation thus reducing
the number of independent free parameters controlling the system, thus is employed here.
Using the defined nondimensional variables listed in Table 1, Eqs. (1)-(16) become the
following equations in the dimensionless forms as:
$\alpha_{Dri}\left(\dfrac{\partial^2 s_{Di}}{\partial r_D^2}+\dfrac{1}{r_D}\dfrac{\partial s_{Di}}{\partial r_D}\right)+\alpha_{Dzi}\dfrac{\partial^2 s_{Di}}{\partial z_D^2}=\dfrac{\partial s_{Di}}{\partial t_D}$   (17)
$s_{Di}\left(r_D,z_D,0\right)=0$    (18)
$s_{Di}\left(\infty,z_D,t_D\right)=0$    (19)
$\displaystyle\lim_{r\to 0}r_D\,\frac{\partial s_{1D}}{\partial r_D}=\begin{cases}0 & l_D<z_D\le 1\\[2mm]-2\dfrac{Q_D(t_D)}{l_D-d_D} & d_D\le z_D\le l_D\\[2mm]0 & 0\le z_D<d_D\end{cases}$    (20)
$Q\left(t_D\right)=1+\left(Q_{1D}-1\right)e^{-\alpha_D t_D}$    (21)
$\displaystyle\lim_{r_D\to 0}r_D\,\frac{\partial s_{D2}}{\partial r_D}=0$    (22)
$\displaystyle\lim_{r_D\to 0}r_D\,\frac{\partial s_{D3}}{\partial r_D}=0$    (23)
$s_{D1}\left(r_D,z_D,t_D\right)=s_{D2}\left(r_D,z_D,t_D\right),\quad z_D=1$    (24)
$\dfrac{\partial s_{D1}\left(r_D,z_D,t_D\right)}{\partial z_D}=\kappa_1\dfrac{\partial s_{D2}\left(r_D,z_D,t_D\right)}{\partial z_D},\quad z_D=1$    (25)
$s_{D1}\left(r_D,z_D,t_D\right)=s_{D3}\left(r_D,z_D,t_D\right),\quad z_D=0$    (26)
$\dfrac{\partial s_{D1}\left(r_D,z_D,t_D\right)}{\partial z_D}=\kappa_2\dfrac{\partial s_{D3}\left(r_D,z_D,t_D\right)}{\partial z_D},\quad z_D=0$    (27)
Case 1,
$s_{D2}\left(r_D,z_D,t_D\right)=0,\quad z_D=B_{D2}$    (28)
$s_{D3}\left(r_D,z_D,t_D\right)=0,\quad z=-B_{D3}$    (29)
Case 2,



$\dfrac{\partial s_{D2}(r_D,z_D,t_D)}{\partial z}=0,\ z_D=B_{D2}$    (30)
$\dfrac{\partial s_{D3}(r_D,z_D,t_D)}{\partial z_D}=0,\ z_D=-B_{D3}$    (31)
Case 3,
$s_{D2}(r_D,z_D,t_D)=0,\ z_D=B_{D2}$    (32)
$\dfrac{\partial s_{D3}(r_D,z_D,t_D)}{\partial z_D}=0,\ z_D=-B_{D3}$    (33)
in which the subscript '$D$' designates nondimensional terms.
**2.2.2 Dimensionless solutions for Case 1**
With the help of the constant-head boundary at the top and bottom expressed in Eqs. (28)
and (29), the drawdown solutions in the three layers can be derived by performing
Laplace-Hankel transform, the detailed derivations are shown in Appendix A.
The dimensionless drawdown for the middle-pumped layer in Laplace space yields
$\bar{s}_{D1}=\int_0^{\infty}\left\{\hat{\bar{u}}_D(\lambda,z_D,p)-4\left[\hat{\bar{u}}(r_D,0,p)\gamma_2 f_{11}+\hat{\bar{u}}(r_D,1,p)\gamma_1 f_{12}\right]/\chi_1\right\}\lambda J_0(\lambda r_D)d\lambda$    (34a)
where
$\hat{\bar{u}}_D(\lambda,z_D,p)=2\dfrac{\cosh(\xi_1\zeta_D)-\delta\hat{\bar{u}}_D(\xi_1,z_D)}{\alpha_{Dz1}\xi_1^2(l_D-d_D)}$    (34b)
$\zeta_D=\begin{cases}z_D-l_D & l_D<z_D\le 1\\ 0 & d_D\le z_D\le l_D\\ d_D-z_D & 0\le z_D<d_D\end{cases}$    (34c)
$\delta\hat{\bar{u}}_D(\xi_1,z_D)=\dfrac{\sin\left[\xi_1(1-l_D)\right]\cosh(\xi_1 z_D)+\cosh\left[\xi_1(1-z_D)\right]\sinh(\xi_1 d_D)}{\sinh(\xi_1)}$    (34d)
$f_{11}=\sinh\left[\xi_1(1-z_D)\right]\left(\cosh\theta_1+\cosh\theta_2\right)\gamma_1+\cosh\left[\xi_1(1-z_D)\right]\left(\sinh\theta_1+\sinh\theta_2\right)$    (34e)
$f_{12}=\sinh(\xi_1 z_D)\left(\cosh\theta_1+\cosh\theta_2\right)\gamma_2+\cosh(\xi_1 z_D)\left(\sinh\theta_1-\sinh\theta_2\right)$    (34f)
$\chi_1=2(1+\gamma_1)(1+\gamma_2)\sinh(\xi_1+\theta_1)+2(1-\gamma_1)(1-\gamma_2)\sinh(\xi_1-\theta_1)$
$-2(1+\gamma_1)(1-\gamma_2)\sinh(\xi_1+\theta_2)-2(1-\gamma_1)(1+\gamma_2)\sinh(\xi_1-\theta_2)$    (34g)

in which $J_0(\cdot)$ represents the zero-order and first kind Bessel function, $p$ and $\lambda$ refer,




respectively, to the variables of the transformations of Laplace and Hankel, and, accordingly,
over bar and over hat sign indicate , respectively, the Laplace and Hankel domain parameter,
$\hat{\bar{u}}_D$  provided by Feng et al. (2019) indicates the Hantush (1964) solution in Laplace-Hankel
domain for a partially penetration well with variable discharge in a single confined aquifer.
The dimensionless solution of drawdown in the upper unpumped layer yields
$$\bar{s}_{D2} = 8\int_0^\infty \frac{\sinh\left[\xi_2\left(B_{D2}-z_D\right)\right]}{\chi_1}\cosh\left(\xi_3 B_{D3}\right)\left\{\hat{\bar{u}}\left(r_D,0,p\right)\gamma_2 - \hat{\bar{u}}\left(r_D,1,p\right)\left[\gamma_2\cosh\left(\xi_1\right)+\sinh\left(\xi_1\right)\right]\right\}\lambda J_0\left(\lambda r_D\right) \quad (35)$$
The semi-analytical solution of dimensionless drawdown in the lower unpumped layer is
written as
$$\bar{s}_{D3} = 8\int_0^\infty \frac{\sinh\left[\xi_3\left(B_{D3}+z_D\right)\right]}{\chi_1}\left\{\hat{\bar{u}}\left(r_D,0,p\right)g_{31} - \hat{\bar{u}}\left(r_D,1,p\right)\gamma_1\cosh\left[\xi_2\left(B_{D2}-1\right)\right]\right\}\lambda J_0\left(\lambda r_D\right)d\lambda \quad (36a)$$
where
$$g_{31} = \gamma_1\cosh\left[\xi_2\left(B_{D2}-1\right)\right]\cosh\xi_1 + \sinh\left[\xi_2\left(B_{D2}-1\right)\right]\sinh\xi_1 \quad (36b)$$
**2.2.3 Dimensionless solutions for Case 2**
If the boundaries at the top and bottom of the aquifer system satisfy the no-flux
boundary written in Eqs.(30)-(31), one can follow the procedures listed in Appendix A and
develop the semi-analytical solutions of dimensionless drawdown in individual layer of the
three-layer aquifer system. The drawdown solution in Laplace-domain in the middle-pumped
layer yields
$$\hat{\bar{s}}_{D1} = \int_0^\infty \left\{\hat{\bar{u}}_D\left(\lambda,z_D,p\right) + 4\left[\hat{\bar{u}}\left(r_D,0,p\right)\gamma_2 f_{21} + \hat{\bar{u}}\left(r_D,1,p\right)\gamma_1 f_{22}\right]/\chi_2\right\}\lambda J_0\left(\lambda r_D\right)d\lambda \quad (37a)$$
where
$$f_{21} = -\sinh\left[\xi_1\left(1-z_D\right)\right]\left(\cosh\theta_2-\cosh\theta_1\right)\gamma_1 + \cosh\left[\xi_1\left(1-z_D\right)\right]\left(\sinh\theta_1-\sinh\theta_2\right) \quad (37b)$$
$$f_{22} = \sinh\left(\xi_1 z_D\right)\left(\cosh\theta_1-\cosh\theta_2\right)\gamma_2 + \cosh\left(\xi_1 z_D\right)\left(\sinh\theta_1+\sinh\theta_2\right) \quad (37c)$$





$$\chi_2 = -2(1+\gamma_1)(1+\gamma_2)\sinh(\xi_1+\theta_1) - 2(1-\gamma_1)(1-\gamma_2)\sinh(\xi_1-\theta_1)$$
$$-2(1+\gamma_1)(1-\gamma_2)\sinh(\xi_1+\theta_2) - 2(1-\gamma_1)(1+\gamma_2)\sinh(\xi_1-\theta_2) \quad (37d)$$

The drawdown solution in Laplace domain in the upper unpumped layer yields
$$\bar{s}_{D2} = 8\int_0^\infty \frac{\cosh[\xi_2(B_{D2}-z_D)]}{\chi_2}\Big[\gamma_2\sinh(\xi_3 B_{D3})\hat{\bar{u}}(r_D,0,p) - \hat{\bar{u}}(r_D,1,p)M\Big]\lambda J_0(\lambda r_D) \quad (38)$$
in which  $M = \gamma_2\sinh(\xi_3 B_{D3})\cosh(\xi_1) + \cos(\xi_3 B_{D3})\sinh(\xi_1)$ .
The drawdown solution in Laplace domain in the lower unpumped layer can be
expressed as
$$\bar{s}_{D3} = 8\int_0^\infty \frac{\cosh[\xi_3(B_{D3}+z_D)]}{\chi_2}\Big\{-\hat{\bar{u}}(r_D,0,p)g_{32} + \hat{\bar{u}}(r_D,1,p)\gamma_1\sinh[\xi_2(B_{D2}-1)]\Big\}\lambda J_0(\lambda r_D) \quad (39a)$$
where
$$g_{32} = \gamma_1\sinh[\xi_2(B_{D2}-1)]\cosh\xi_1 + \cosh[\xi_2(B_{D2}-1)]\sinh\xi_1 \quad (39b)$$
**2.2.4 Dimensionless solutions for Case 3**
By analogy, with the use of the constant-head boundary at the top and the no-flux
boundary at the bottom, which are, respectively, described by Eq. (32) and Eq. (33), one can
develop the nondimensional drawdown solutions in Laplace space for the middle (pumped)
layer as:
$$\hat{\bar{s}}_{D1} = \hat{\bar{u}}_D(\lambda, z_D, p) + \frac{4}{\chi_3}\Big[\hat{\bar{u}}(r_D,0,p)\gamma_2 f_{31} + \hat{\bar{u}}(r_D,1,p)\gamma_1 f_{32}\Big] \quad (40a)$$
where
$$f_{31} = -\sinh[\xi_1(1-z_D)](\sinh\theta_2 - \sinh\theta_1)\gamma_1 + \cosh[\xi_1(1-z_D)](\sinh\theta_1 - \sinh\theta_2) \quad (40b)$$
$$f_{32} = \sinh(\xi_1 z_D)(\sinh\theta_1 - \sinh\theta_2)\gamma_2 + \cosh(\xi_1 z_D)(\cosh\theta_1 + \cosh\theta_2) \quad (40c)$$
$$\chi_3 = -2(1+\gamma_1)(1+\gamma_2)\cosh(\xi_1+\theta_1) + 2(1-\gamma_1)(1-\gamma_2)\cosh(\xi_1-\theta_1)$$
$$-2(1+\gamma_1)(1-\gamma_2)\cosh(\xi_1+\theta_2) + 2(1-\gamma_1)(1+\gamma_2)\cosh(\xi_1-\theta_2) \quad (40d)$$

and, for the upper unpumped layer, one has





$$\bar{s}_{D2} = 8\int_0^\infty \frac{\sinh\left[\xi_2\left(B_{D2}-z_D\right)\right]}{\chi_3}\left[\gamma_2 \sinh\left(\xi_3 B_{D3}\right)\hat{\bar{u}}\left(r_D,0,p\right)-\hat{\bar{u}}\left(r_D,1,p\right)N\right]\lambda J_0\left(\lambda r_D\right) \quad (41)$$
in which   $N = \gamma_2 \sinh(\xi_3 B_{D3})\cosh(\xi_1) + \cosh(\xi_3 B_{D3})\sinh(\xi_1)$ .
and, for the lower pumped layer, one has
$$\bar{s}_{D3} = 8\int_0^\infty \frac{\cosh\left[\xi_3\left(B_{D3}+z_D\right)\right]}{\chi_3}\left\{-\hat{\bar{u}}\left(r_D,0,p\right)g_{33}+\hat{\bar{u}}\left(r_D,1,p\right)\gamma_1 \cosh\left[\xi_2\left(B_{D2}-1\right)\right]\right\}\lambda J_0\left(\lambda r_D\right) \quad (42a)$$
where
$$g_{33} = \gamma_1 \cos\left[\xi_2\left(B_{D2}-1\right)\right]\cosh\xi_1 + \sinh\left[\xi_2\left(B_{D2}-1\right)\right]\sinh\xi_1 \quad (42b)$$
**2.3 Special cases**
**2.3.1 Special cases in a three-layer aquifer**
If removing the effect of the radial flow in the upper and lower unpumped layer ($K_{r2}$ =
$K_{r3}$ = 0,  $\alpha_{r2} = \alpha_{Dr2} = 0$ ,  $\alpha_{r3} = \alpha_{Dr3} = 0$ ,  $\xi_2^2 = p / \alpha_{Dz2}$  and  $\xi_3^2 = p / \alpha_{Dz3}$ ), the developed solutions
of Eqs. (33) – (41) agree with the solutions for a conventional aquitard-aquifer-aquitard
system with the assumption of only considering the vertical flows in the unpumped layers, as
in previous works of Hantush (1960), Moench (1985) and Chen et al. (2020). The condition
for this assumption is that the permeability of the middle-pumped aquifer is usually larger at
least two orders of magnitude than that of the upper and lower aquitards.
Additionally, the transient dimensionless solutions in the three-layer aquifer system
caused by a partially penetrating constant-rate pumping well in the middle layer can be
obtained from Eqs. (34) – (42) by setting $Q_{1D}$ = 1, and as far as the author knows, these
solutions have not been developed in the existing studies.
**2.3.2 Special cases in a two-layer aquifer**
If the lower unpumped layer is absence, one has $B_{D3}$ = 0,  $\gamma_2 = 0$, and $\theta_1 = \theta_2 = \xi_2\left(B_{D2}-1\right)$,



the dimensionless drawdown solutions in a two-layer aquifer having a constant-head and
no-flow boundary at the top (Case 2 and Case3) can be, respectively, developed from Eqs.
(37) – (42) and the detailed expression can be, respectively, given by:
Case 2: for the pumped layer, one has
$\hat{\bar{s}}_{D1} = \hat{\bar{u}}_D(\lambda, z_D, p) + 2\dfrac{\hat{\bar{u}}(r_D, 1, p)}{\chi_2'}\gamma_1\cosh(\xi_1 z_D)\sinh[\xi_2(B_{D2}-1)]$    (43)
and for the upper unpumped layer, one has
$\hat{\bar{s}}_{D2} = -2\dfrac{\hat{\bar{u}}(r_D, 1, p)}{\chi_2'}\cosh[\xi_2(B_{D2}-z_D)]\sinh(\xi_1)$    (44)
with
$\chi_2' = (\gamma_1-1)\sinh[\xi_1-\xi_2(B_{D2}-1)]-(\gamma_1+1)\sinh[\xi_1+\xi_2(B_{D2}-1)]$    (45)
Case 3: for the pumped layer, one has
$\hat{\bar{s}}_{D1} = \hat{\bar{u}}_D(\lambda, z_D, p) + 2\dfrac{\hat{\bar{u}}(r_D, 1, p)}{\chi_1}\gamma_1\cosh[\xi_2(B_{D2}-1)]\cosh(\xi_1 z_D)$    (46)
and for the upper unpumped layer, one has
$\hat{\bar{s}}_{D2} = -2\dfrac{\hat{\bar{u}}(r_D, 1, p)}{\chi_3'}\sinh(\xi_1)\sinh[\xi_2(B_{D2}-z_D)]$    (47)
with
$\chi_3' = (1-\gamma_1)\cosh[\xi_1-\xi_2(B_{D2}-1)]-(1+\gamma_1)\cosh[\xi_1+\xi_2(B_{D2}-1)]$    (48)
These solutions of drawdown agree with the solutions of Feng et al. (2019), describing
flow in a two-layer aquifer system pumped by a partial penetration well of a variable/constant
discharge subject to a zero-drawdown and no-flux conditions at the top boundary.
Further, if $Q_{1D} = 1$,  $\alpha_{r2} = \alpha_{Dr2} = 0$  and  $\xi_2^2 = p/\alpha_{Dz2}$, the drawdown solutions of Eqs. (43)
– (45) are equal to the solutions having different expressions developed by Feng and Zhan
(2015), that can be applied to investigate the drawdown caused by a pumping well of partial





penetration in an aquitard-aquifer system where the horizontal flow in the upper layer is
neglected and a zero-drawdown condition can be imposed at the top boundary.
**2.3.3 Special cases in a single-layer aquifer**

If ignoring the leakage effect between two adjacent layers, the present pumped layer

drawdown solutions can reduce to the solution of Hantush (1964) for flow in a confined
aquifer due to a partially penetrated well with constant pumping rate ($Q_{1D} = 1$).    When the
pumped layer is fully penetrated by a well with an exponentially decreasing discharge and
leakage is not considered, Eqs. (34b)–(34d) collapse to the drawdown solution of Wen et al.
(2017). Additionally, the classical solution of Theis is also included in the new obtained
solution when $Q_{1D} = 1$.
**2.4 Numerical inversion of the solutions**

So far, the Laplace-domain solutions of nondimensional drawdown for diverse cases are

developed. In this study, a numerical integration algorithm (Ogata, 2005) with the method
using the zeros of the Bessel functions as nodes can be performed to calculate the infinite
integral associated with the transformation of Hankel, and the method of de Hoog algorithm
(De Hoog et al., 1982) is able to be applied to solve the transformation of Laplace. Finally,
one can obtain the solutions in time domain by successively using the two method of
numerical inversion of Hankel transform and Laplace transform respectively. The verification
and validation of the method have been proven and more details can be found in the study of
Feng et al. (2020) and Liang et al. (2018), which is not discussed herein.
**3 Results**

The dimensionless drawdown response due to a partial penetration well pumped at an




exponentially decreasing discharge is explored in the following from a number of
perspectives. Default values for realistic aquifers are used in the following analysis: $B_1 = 20$m;
$B_2 = 30$m; $B_3 = 10$m; $K_{r1} = K_{z1} = 10^{-4}$ ms$^{-1}$; $K_{r2} = K_{z2} = 10^{-6}$ ms$^{-1}$; $K_{r3} = K_{z3} = 10^{-6}$ ms$^{-1}$; $S_{s1} =$
$2\times10^{-5}$ m$^{-1}$ ; $S_{s2} = 10^{-3}$ m$^{-1}$ ; $S_{s3} = 10^{-6}$ m$^{-1}$; $Q_1 = 0.005$m$^3$s$^{-1}$; $Q = 0.002$m$^3$s$^{-1}$. One can see that
the upper and lower unpumped layers have the same hydraulic properties of aquitard
composed of clay soil for simplicity, and middle pumped layer may be composed of sand
soils in reality. Under this circumstance, the three-layer system becomes a commonly
investigated three-layer aquitard-aquifer-aquitard system (Hantush, 1960; Moench, 1985;
Wen et al, 2011; Chen et al., 2020), which will be analyzed for comparison with existing
works, though the presented solution applies to a general three-layer aquifer systems with no
restrictions on the hydraulic parameter (e.g. permeability, specific storage) and the thickness
of each layer. Aquifer anisotropy and different permeability contrasts among individual layers
will also be explored to show the importance of considering both vertical and horizontal
flows for each of the three layers, no matter the layer is pumped or unpumped.
**3.1 Comparison with available solutions**

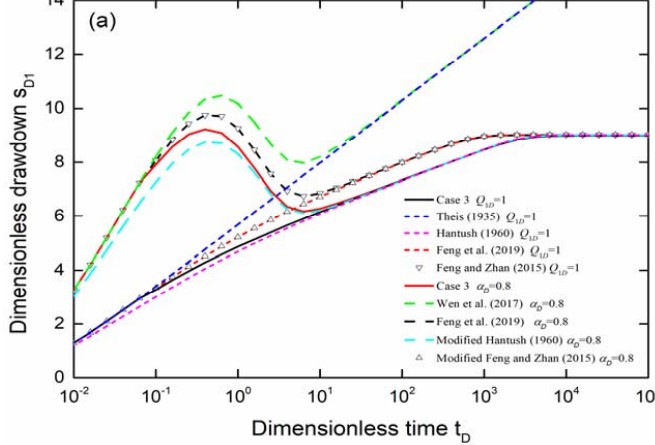




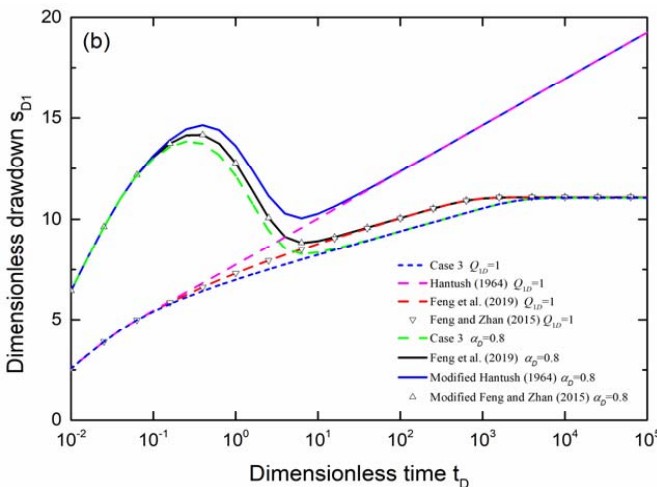


Fig.2 Comparison of the type curves for pumped layer provided by the newly developed
solution for Case 3 and other existing solutions for a full penetration well (a) and a partial
penetration well (b) with $r_D = 0.1$, $z_D = 0.5$, $\kappa_1 = \kappa_2 = 10^{-2}$, $\alpha_{Dz2} = \alpha_{Dr2} = 2 \times 10^{-4}$, $\alpha_{Dz3} = \alpha_{Dr3} =$
$2 \times 10^{-4}$, $\alpha_D = 0.8$, $Q_{1D} = 2.5$, $B_{D2} = 1.5$, $B_{D3} = 0.5$.

Fig. 2 (a) illustrates the drawdown responses of the pumped layer at $r_D = 0.1$ and $z_D =$
0.5 caused by a full penetration pumping well ($l_D = 1$, $d_D = 0$) in an aquitard-aquifer-aquitard
system (Case 3 in this study, Hantush, 1960), an aquitard-aquifer system (Feng and Zhan,
2015, Feng et al. 2019), and a confined aquifer system (Theis, 1935, Wen et al., 2017). Fig. 2
(b) shows the pumped aquifer drawdown at the same location as Fig. 2 (a) due to a partial
penetration pumping well ($l_D = 0.75$, $d_D = 0.25$) in present solution for Case 3, solutions of
Feng and Zhan (2015) and Feng et al. (2019) for a leaky confined aquifer system, and
Hantush (1964) for a nonleaky-confined aquifer system. Both the cases of constant ($Q_{1D} = 1$)
and variable discharge ($Q_{1D} = 2.5$, $\alpha_D = 0.8$) are taken into account in this figure.
No matter what the well discharge is, under the circumstance of a full penetration well,
the early-time drawdown for almost all study agree with one another except for the (modified)
Hantush (1960) solution. The results are slightly larger than that of (modified) Hantush (1960)
for an aquitard-aquifer-aquitard system if using the Hantush-Jacob approximation and the
assumption of only considering the radial flow in the pumped layer and vertical flow in the
unpumped aquitard. Because the leakage effect is regarded as a sink/source term introduced
in the pumped aquifer governing equation in Hantush (1960), it is no strange to see a smaller
drawdown in early time, as demonstrated in Fig. 2. The drawdown of Theis (1935) and Wen
et al. (2017) with a full penetration well in Fig. 2 (a) or Hantush (1964) with a partial
penetration well in Fig. 2 (b) is always larger than the others with the increasing of pumping
time due to no leakage from adjacent layers. The intermediate time-drawdown in a leaky
confined aquifer is greater than that in an aquitard-aquifer-aquitard system, which may be
caused by less leakage into the pumped aquifer derived entirely from the upper aquitard
storage. The late-time steady-state drawdowns can be found in two-layer and three-layer
aquifer system and their values are almost the same as each other. Moreover, the time to
approach the steady state for two-layer aquifer system (Feng and Zhan, 2015, Feng et al.,
2019) is much earlier than that for three-layer aquifer system (Hantush, 1960, present study
for Case 3), this is to be understood that the water from top boundary of the aquifer system of
two-layer is also much quicker to supply the pumped aquifer because the pumped aquifer
drawdown is not influenced by the storage of the lower layer in the aquifer system of
three-layer.

Comparison of the dimensionless drawdown solution induced by a full penetration

pumping well obtained by this study for Case 3 and (modified) Hantush (1960), one can only
see the difference at early and intermediate times when $t_D$ is smaller than about $10^2$, as


demonstrated in Fig. 2 (a). This can be attributed to the following aspects. Firstly, the
Hantush-Jacob approximation is used in (modified) Hantush (1960). Secondly, the flow in the
radial direction of aquitard and flow in the vertical direction of the pumped aquifer are not
taken into consideration in (modified) Hantush (1960). However, the present study takes
account of the horizontal and vertical flows in each layer, as we as treat the leakage across the
two adjacent layers as continuity boundary conditions rather than a simplified volumetric
sink/source term, accordingly, our general analytical model can reflect the actual leakage
process. Therefore, one can conclude that the use of the Hantush-Jacob approximation should
be deliberated, especially at the early pumping time for a fully penetrating well. One can see
from Fig. 2 (b) that the storage of lower unpumped aquitard primarily affects the drawdown
distribution for the three-layer aquifer system of Case 3 at the intermediate pumping time,
signifying that the hydraulic parameters of lower aquitard can be estimated by using the
observed data at this stage. In additional, more comparative analysis for the pumped aquifer
drawdown in a confined aquifer with a pumping well of full penetration (Theis, 1935, Wen et
al. (2017) or of partial penetration (Hautush, 1964) and in a two-layer aquifer with a
full/partial penetration well (Feng and Zhan, 2015, Feng et al., 2019) can be found in the
work of Feng et al. (2019), which is not repeated herein.

It should be remarked that the typical curves of drawdown versus pumping time have

two inflection points during the decaying period of pumping rate, and more discussion and
explanation for this feature can be found in Wen et al. (2017). At last, one can see from Fig. 2
(a) in comparison with Fig. 2 (b) that the pumped layer drawdown due to a partial penetration
pumping well is greater than that a full penetration pumping well at the same value of





pumping time, indicating that the effect of well partial penetration needs to be considered.
**3.2 Effect of various top and bottom boundaries**

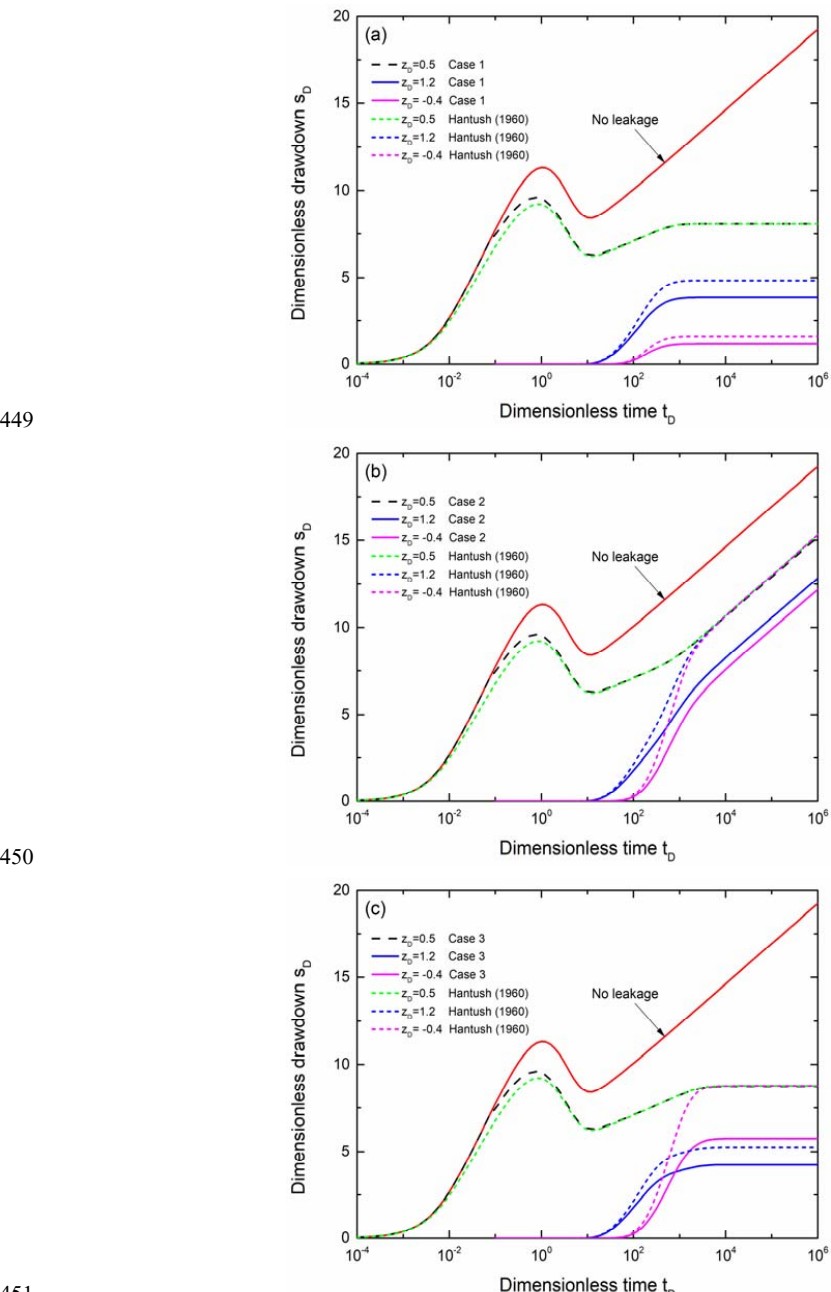




Fig.3 The typical curves of dimensionless drawdown versus dimension time in the pumped





layer and unpumped layers under different top and bottom boundary (a) for Case1 (b) for
Case 2 and (c) for Case 3 with $r_D = 0.1$, $z_D = 0.5$, $l_D = 1.0$, $d_D = 0$, $\kappa_1 = \kappa_2 = 10^{-2}$, $\alpha_{Dz2} = \alpha_{Dr2} = $
$2 \times 10^{-4}$, $\alpha_{Dz3} = \alpha_{Dr3} = 2 \times 10^{-4}$, $\alpha_D = 0.8$, $Q_{1D} = 2.5$, $B_{D2} = 1.5$, $B_{D3} = 0.5$.

Fig. 3 shows the changes of drawdown at $r_D = 0.1$ in the middle pumped layer ($z_D = 0.5$),

in the upper layer ($z_D = 1.2$) and in the lower unpumped layer ($z_D = -0.4$) for Case 1 (a), Case
2 (b), and Case 3 (c) under the condition of a well of full penetration ($l_D = 1$, $d_D = 0$). The
solution of Hantush (1960) is included in this figure for comparison purposes and the case of
no leakage (Wen et al., 2017) is also considered as a reference. The curves of drawdown
versus time for the pumped layer obtained by this study and Hantush (1960) have almost the
same feature during the entire pumping stage and their deviations are mainly occurred at the
stage of $10^{-2} < t_D < 10^1$, as illustrated in the subgraphs of Fig.3 with three different cases.
Additionally, as for the drawdown response in the two unpumped layers, one can find from
Fig.3 that the drawdown developed by this study is always larger than that of Hantush (1960)
as the pumping time goes by and a relatively stable error between them can be found at late
time. This is due to fact that the influence of radial flow in the unpumped layer is ignored by
Hantush (1960). What is more, Fig. 3 (b) and Fig. 3 (c) demonstrate that the drawdown for
the lower unpumped layer is nearly identical to that for the pumped layer if only taking
account of the vertical flow in the unpumped layer. In other words, whether the radial flow in
the unpumped layer is overlooked or not, one can see that from the comparison of drawdowns
in the pumped layer with that in the unpumped layer for Case 2 and Case 3.

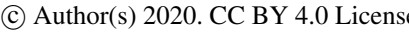


Fig.4 Comparison of the typical curves of dimensionless drawdown versus dimension time in

the pumped layer and unpumped layers under diverse cases with $r_D = 0.1$, $z_D = 0.5$, $l_D = 0.75$,

$d_D = 0.25$, $\kappa_1 = \kappa_2 = 10^{-2}$, $\alpha_{Dz2} = \alpha_{Dr2} = 2 \times 10^{-4}$, $\alpha_{Dz3} = \alpha_{Dr3} = 2 \times 10^{-4}$, $\alpha_D = 0.8$, $Q_{1D} = 2.5$, $B_{D2}$

$= 1.5$, $B_{D3} = 0.5$.


In order to compare the drawdowns under different boundaries at the top and bottom of

the aquifer system, Fig. 4 displays the drawdown changes at $r_D = 0.1$ in the pumped layer ($z_D$
$= 0.5$) and in the unpumped layers ($z_D = 1.2$ and $z_D = -0.4$) for all three cases with a partial
penetration pumping well ($l_D = 0.75$, $d_D = 0.25$). Notably, the no leaky case (modified,
Hantush, 1964) is plotted as a reference in this figure. Fig. 4 shows that the influence of the
type of top and boundary can be ignored in exploring drawdown at the early and intermediate
pumping time, however, its influence on the late-time drawdown behavior is obvious, and
one can find that the drawdowns for Cases 1 and 3 reach steady state at late pumping stage
because of the unlimited water supply stemmed from the top zero-drawdown boundary. In
addition, the late-time drawdown for Case 3 is greater than that for Case 1 and smaller than





that for Case 2. This is due to the fact that the constant-head boundary at the top and bottom
in Case 1 can give steady and unlimited supply of water, thus leading to the smallest
drawdown among three cases. In another aspect, the no-flux top and bottom boundaries in
Case 2 cannot furnish any supply of water, thus the largest drawdown can be seen among
three cases in this figure.
Fig. 4 also illustrates that the drawdown for Case 2 increases indefinitely with pumping
time and finally parallels with that of the no leakage case. This is caused by the no-flow
boundary at the top and bottom. Furthermore, one cannot see the inflection point of the type
curves for the unpumped layer, indicating that the influence of variable discharge mainly
affects the pumped layer drawdown. This is because the drawdown response for the
unpumped layer appears nearly at the end of the intermediate time and the influence of
variable discharge is very small and can be neglected at this stage, thus the inflection point
cannot be found.

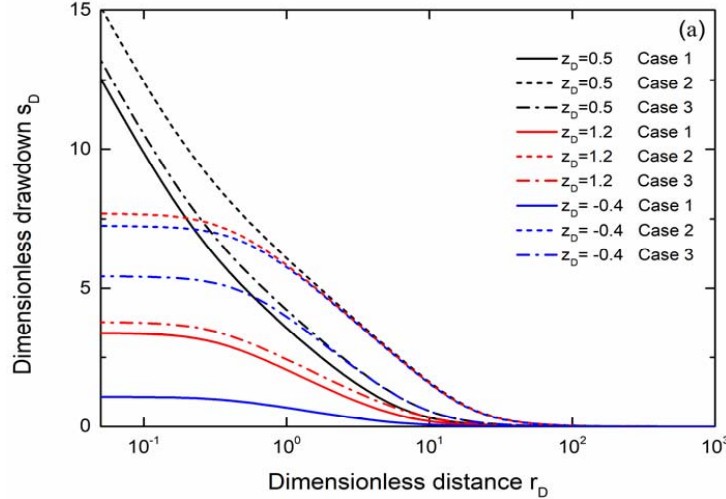


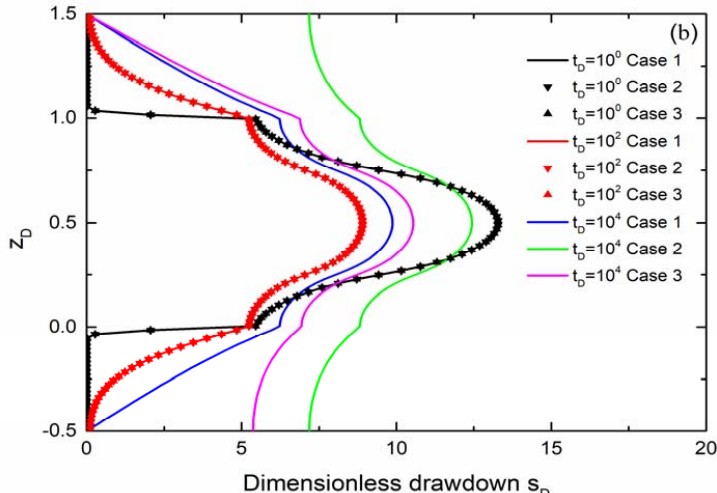


Fig.5 Comparison of the nondimensional drawdown behavior in the pumped layer and

unpumped layers under diverse cases (a) the curves for $s_D$ VS $r_D$ at $t_D = 10^4$, (b) the curves for

$s_D$ VS $z_D$ at $r_D = 0.1$ with $l_D = 0.75$, $d_D = 0.25$, $\kappa_1 = \kappa_2 = 10^{-2}$, $\alpha_{Dz2} = \alpha_{Dr2} = 2 \times 10^{-4}$, $\alpha_{Dz3} = \alpha_{Dr3}$

$= 2 \times 10^{-4}$, $\alpha_D = 0.8$, $Q_{1D} = 2.5$, $B_{D2} = 1.5$, $B_{D3} = 0.5$.


To further investigate the influence of various top and bottom boundaries on drawdown,
Fig. 5 is plotted to demonstrate the drawdown responses in all layers using typical curves of
(a) $s_D$ versus $r_D$ ($z_D$ = 0.5, 1.2 and -0.4 at $t_D$ =10$^4$; (b) $s_D$ versus $z_D$ at $r_D$ = 0.1 with a partial
penetration pumping well ($l_D = 0.75$, $d_D = 0.25$). Fig. 5 (a) shows that the late-time drawdown
at any radial distance $r_D$ for Case 3 is greater than that for Case 1 and smaller than that for
Case 2, and so does the pumping induced influence of the range for different cases, which is
according with the above analysis of drawdown illustrated in Fig.4. It is interesting to find
from Fig. 5 (a) that the drawdown in the pumped layer is nearly the same as that in the lower
unpumped layer for Case 3 at $r_D > 10$, and the same phenomenon can be observed from Fig. 5
(a) for the drawdowns of Case 3 in the two unpumped layers and pumped layer for Case 3 if
$r_D > 40$.
Additionally, the drawdowns along the vertical direction in whole aquifer system under
various top and bottom boundaries are shown in Fig. 5 (b). To clarify, the pumping well of
partial penetration is fixed in the middle of the pumped layer having a screen length of 0.5. It
can be found that the drawdowns along the vertical direction for all three cases coincide with
one another at early and intermediate pumping time ($t_D$ = 1 and $10^2$), however, the
discrepancies among them are significant at a relatively late time of pumping ($t_D = 10^4$). An
interesting observation from Fig. 5 (b) can be included that the drawdowns for Case 1 and
Case 2 have symmetry with the axis $z_D = 0.5$ at the entire pumping time, which are caused by
the identical top and bottom boundaries of the two cases and the same thickness and
hydraulic parameters of the unpumped layers. However, the late-time drawdown for Case 3
has no symmetry and the lower layer drawdown is always smaller than that in the upper layer
at correspondingly position of symmetry, this implies that the lower layer drawdown is
influenced in a greater degree by pumping for Case 3. Besides, the largest drawdown at the
axis of symmetry can be seen during the pumping period for all three cases, as expected. In
general, one can conclude from Fig. 5 that the late-time drawdown is always affected by the
type of top and bottom boundaries at any position within the three-layer aquifer system.
Therefore, except for the location of piezometer (*r* and *z*), one had better clarify the types of
top and bottom boundaries, if the late-time drawdown data are used for the estimation of
parameters of the aquifer system of three-layer.
**3.3 Effect of the variable pumping rate**
Firstly, it points out that Case 3 is hereafter used as an example for demonstration
purpose. It would be easy to analyze drawdown for Case 1 and Case 2 in a similar way when





there is a need. One can know through the above analysis that the pumped aquifer drawdown
is mainly influenced by the variable discharge. Fig. 6 shows only the pumped aquifer
drawdown for Case 3 under different $\alpha_D$ at $r_D$ = 0.1, 0.3 and 0.6. Note that $\alpha_D = \infty$ represents
the final constant pumping rate. One can see that the differences among the type curves for
different decay constants can be seen only at intermediate time. A greater $\alpha_D$ implies that the
well discharge declines much faster to reach the final constant pumping rate, resulting in
smaller drawdowns during the intermediate stage. Additionally, the inflection point of the
curve of drawdown versus time near the pumping well is more obvious than that at a distance
further away from the pumping well. This means that the effect of variable discharge
decreases gradually with the increase of the radial distances and eventually disappears
completely at some distances far enough. From previous study of Wen et al. (2017), one can
use the point of inflection appeared at the stage of the declined pumping discharge at
intermediate time to estimate aquifer parameters. Under this circumstances, Fig. 6 suggests
that the observed data of drawdown near the pumping well would be a good choice.

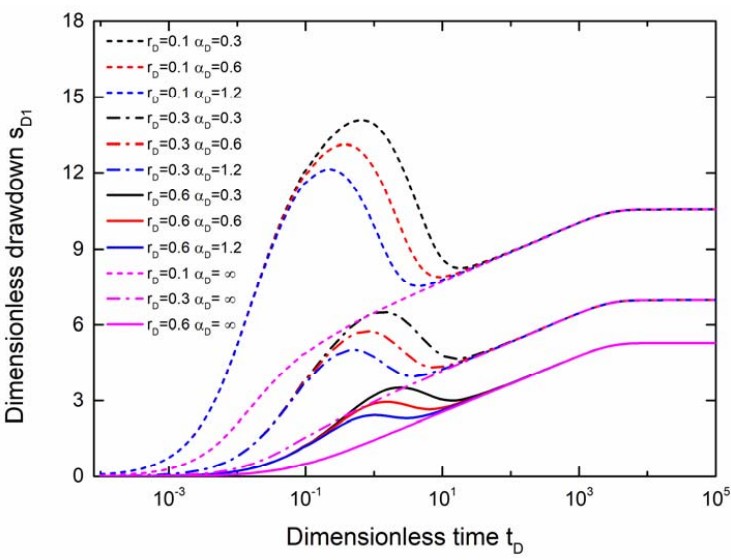



Fig.6 Dimensionless drawdown response in the pumped layer and unpumped layers under
different $\alpha_D$ for Case 3 with $z_D = 0.5$, $l_D = 0.75$, $d_D = 0.25$, $\kappa_1 = \kappa_2 = 10^{-2}$, $\alpha_{Dz2} = \alpha_{Dr2} = 2 \times 10^{-4}$,
$\alpha_{Dz3} = \alpha_{Dr3} = 2 \times 10^{-4}$, $Q_{1D} = 2.5$, $B_{D2} = 1.5$, $B_{D3} = 0.5$.

**3.4 Effect of the unpumped layer thickness**

Fig. 7 shows the drawdown characteristics for the pumped ($z_D = 0.5$) and unpumped

layer ($z_D = 1.1, -0.1$) at $r_D = 0.1$ with a partial penetration well ($l_D = 0.75$, $d_D = 0.25$) for
various unpumped layer thickness ($B_D = B_{D3} = B_{D2} - 1$). Note that the no leakage case (or an
impermeable unpumped layer) is also taken into consideration in this figure for comparison.
The early and intermediate-drawdowns for both pumped aquifer and unpumped layers are not
influenced by the change of the thickness of the unpumped layer, but the larger the thickness
of the unpumped layer, the larger late-time drawdown can be found. In addition, Fig. 7 also
illustrates that the pumped aquifer drawdown is significantly influenced by the leakage from
adjacent layer if compared to the case of no leakage.

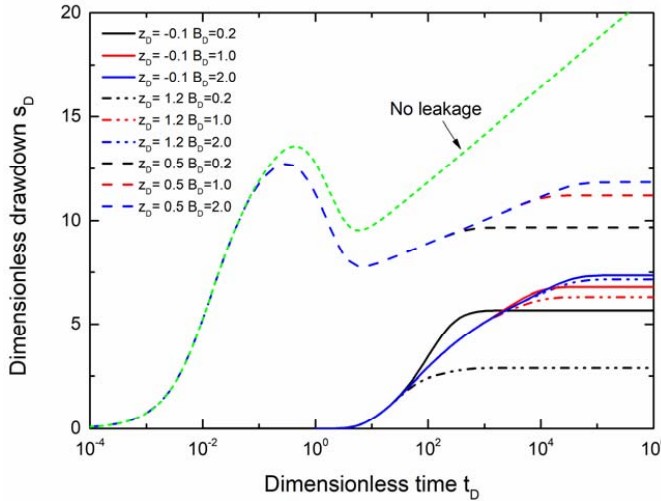


Fig.7 Dimensionless drawdown response in the pumped layer and unpumped layers under



different thickness of the unpumped layers ($B_D = B_{D2} - 1 = B_{D3}$) for Case 3 with $r_D = 0.1$, $z_D =$
0.5, $l_D = 0.75$, $d_D = 0.25$, $\kappa_1 = \kappa_2 = 10^{-2}$, $\alpha_{Dz2} = \alpha_{Dr2} = 2 \times 10^{-4}$, $\alpha_{Dz3} = \alpha_{Dr3} = 2 \times 10^{-4}$, $\alpha_D = 0.8$,
$Q_{1D} = 2.5$, $B_{D2} = 1.5$, $B_{D3} = 0.5$.

**3.5 Effect of anisotropy**

Because of the generality of the established solution, one can easily explore the

influence of anisotropy for each layer on the drawdown in this three-layer system. To be sure,
two schemes of the aquifer system are considered for comparison. The drawdown change in
the classical aquitard-aquifer-aquitard scheme (termed scheme A herein) will show in the
following figures (a), and the drawdown response will also be illustrated in the following
figures (b) for another scheme (termed scheme B herein) of a general aquifer system of
three-layer, having the permeability values of the upper and lower layers being one order of
magnitude smaller (instead of two orders of magnitude smaller as in the default setting) than
that of the middle-pumped layer.

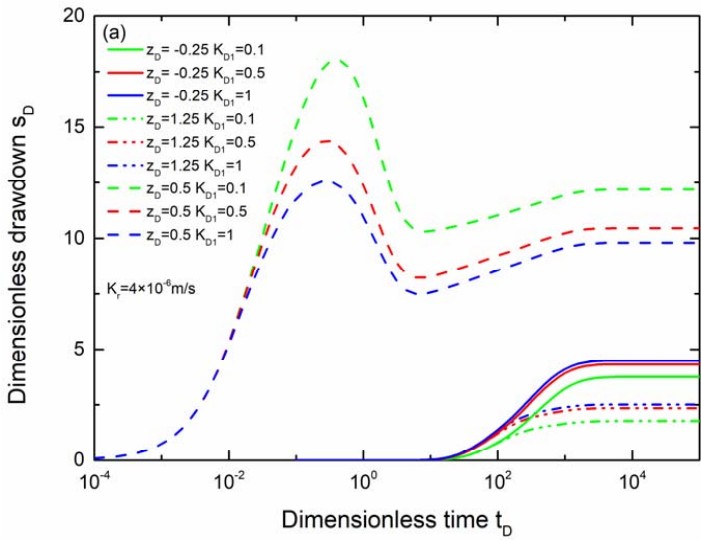




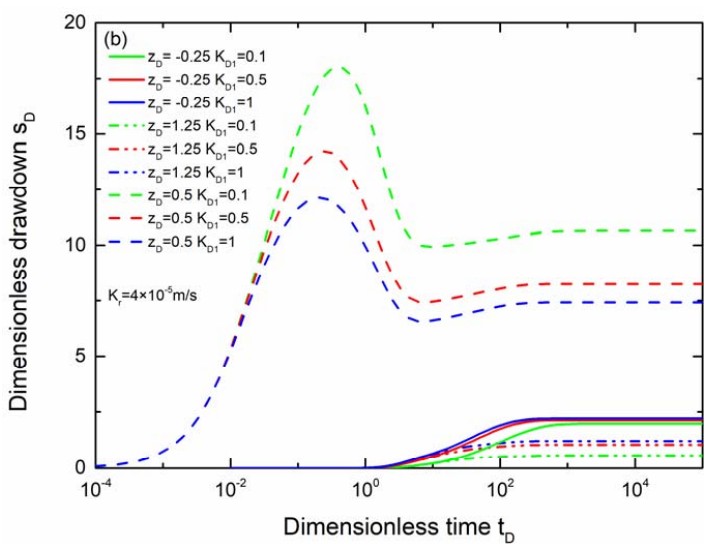


Fig.8 The nondimensional drawdown response in the pumped layer and unpumped layers
under different anisotropy of the pumped layer ($K_{D1} = K_{z1}/K_{r1}$) for Case 3 with $r_D = 0.1$, $\alpha_D =$
0.8, $Q_{1D} = 2.5$, $B_{D2} = 1.5$, $B_{D3} = 0.5$, $l_D = 0.75$, $d_D = 0.25$, $K_{D2} = K_{z2}/K_{r2} = K_{D3} = K_{z3}/K_{r3} = 0.2$,
where (a) $K_r = K_{r2} = K_{r3} = 4\times10^{-6}$m/s, (b) $K_r = K_{r2} = K_{r3} = 4\times10^{-5}$m/s.

Fig. 8 shows the response of drawdown for Case 3 in the pumped layer ($z_D = 0.5$) and in

the upper and lower layers ($z_D = 1.25, -0.25$) at $r_D = 0.1$ with a partial penetration well ($l_D =$
0.75, $d_D = 0.25$) for various anisotropy of the pumped layer ($K_{D1} = K_{z1}/K_{r1}$). Note that $K_{D1} = 1$
refers to the isotropic case, which is included as a reference.

One can see from Fig. 8 that the entire aquifer system for scheme A and scheme B is

affected by the change of the pumped layer anisotropy almost during the entire pumping time.
The pumped layer drawdown decreases with an increase of the anisotropy ratio and a larger
$K_{D1}$ results in larger drawdowns for the upper and lower unpumped layers. Comparing the
drawdowns for scheme A shown in Fig. 8 (a) and for scheme B listed in Fig. 8 (b), one can
see that the drawdown for scheme A is always larger than that for scheme B. This is because





the difference of the permeability of the unpumped layers and pumped layer for scheme B is
not as significant as that for scheme A, and the capacity of water supply of the unpumped
layers for scheme B is much stronger than that for scheme A. Therefore, it is much easier to
obtain the water supply from the top boundary, thus a smaller drawdown is seen as illustrated
in Fig. 8 (b). Overall, the pumped layer anisotropy is of great importance to ascertaining the
drawdown behavior of the entire three-layer aquifer system.

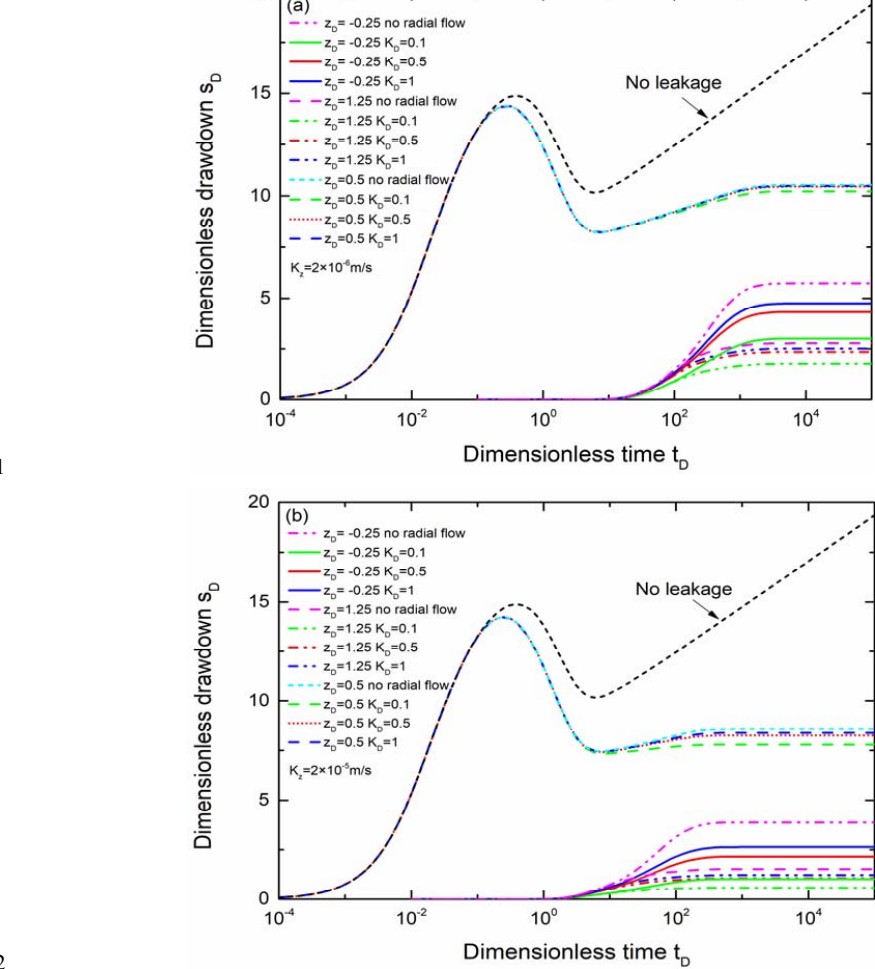



Fig.9 The nondimensional drawdown change in the pumped layer and unpumped layers
under different anisotropy of the unpumped layers ( $K_D = K_{z2}/K_{r2} = K_{z3}/K_{r3}$) for Case 3 with


$r_D = 0.1$, $\alpha_{Dz2} = \alpha_{Dz3} = 2 \times 10^{-4}$, $\alpha_D = 0.8$, $Q_{1D} = 2.5$, $B_{D2} = 1.5$, $B_{D3} = 0.5$, $K_{D1} = K_{z1}/K_{r1} = 0.5$,
$K_{r2} = K_{r3}$, $l_D = 0.75$, $d_D = 0.25$, in which (a) $\kappa_1 = \kappa_2 = 0.04$, $\alpha_{Dr2} = \alpha_{Dr3} = 4 \times 10^{-5}$, $K_z = K_{z2} =$
$K_{z3} = 2\times10^{-6}$m/s and (b) $\kappa_1 = \kappa_2 = 0.4$, $\alpha_{Dr2} = \alpha_{Dr3} = 4 \times 10^{-4}$, $K_z = K_{z2} = K_{z3} = 2\times10^{-5}$m/s.

Fig. 9 demonstrates the drawdown changes for Case 3 in an anisotropic pumped layer
($z_D = 0.5$, $K_{D1} = 0.5$ and $K_{r1} = 10^{-4}\,m/s$) and anisotropic upper and lower layers ($z_D = 1.25$ and
-0.25) for various anisotropy ratios of unpumped layer ($K_D = K_{D2} = K_{z2}\,/\,K_{r2} = K_{D3} = K_{z3}\,/\,K_{r3}$)
at $r_D = 0.1$ with a pumping well of partial penetration ($l_D = 0.75$ and $d_D = 0.25$). It should be
mentioned that the vertical permeability of the unpumped layer is to be kept on hold in Fig. 9,
where (a) $K_z = K_{z2} = K_{z3} = 2\times10^{-6}m/s$ and (b) $K_z = K_{z2} = K_{z3} = 2\times10^{-5}$m/s. The case of an
isotropic unpumped layer ($K_D = 1$) is considered in both subgraphs, and the case of ignoring
the radial flow in unpumped layer is depicted as well for comparison in Fig. 9. One can
obviously see from Fig. 9 that the influence of various anisotropy ratios on the pumped layer
drawdowns almost coincide with the case of the unpumped layer with no horizontal low for
scheme A if $K_D \geq 0.5$. However, when $K_D$ is 0.1 for scheme A, the anisotropy of the
unpumped layers significantly affects the pumped layer drawdown at the late pumping time
as demonstrated in Fig. 9 (a). The influence of the unpumped layers anisotropy on the
pumped layer drawdown for scheme B is more obvious than that for scheme A at
intermediate and late times, it can be seen from Fig. 9 (b). In addition, no matter what the
value of anisotropy $K_D$ is, the change of $K_D$ has an appreciable influence on the unpumped
layer drawdowns for both scheme A and scheme B. Finally, one still can conclude from Fig. 9
that the drawdown for scheme A is generally larger than that for scheme B at the same
position within the aquifer system of three-layer and at the same pumping time. Overall, the





radial and vertical flows in the unpumped layer (effect of anisotropy) should be considered in
determining drawdown responses around the pumping well, especially to the general case
without large contrast of hydraulic conductivity among the unpumped layers and the pumped
layer.
**3.6 The effect of well partial penetration**

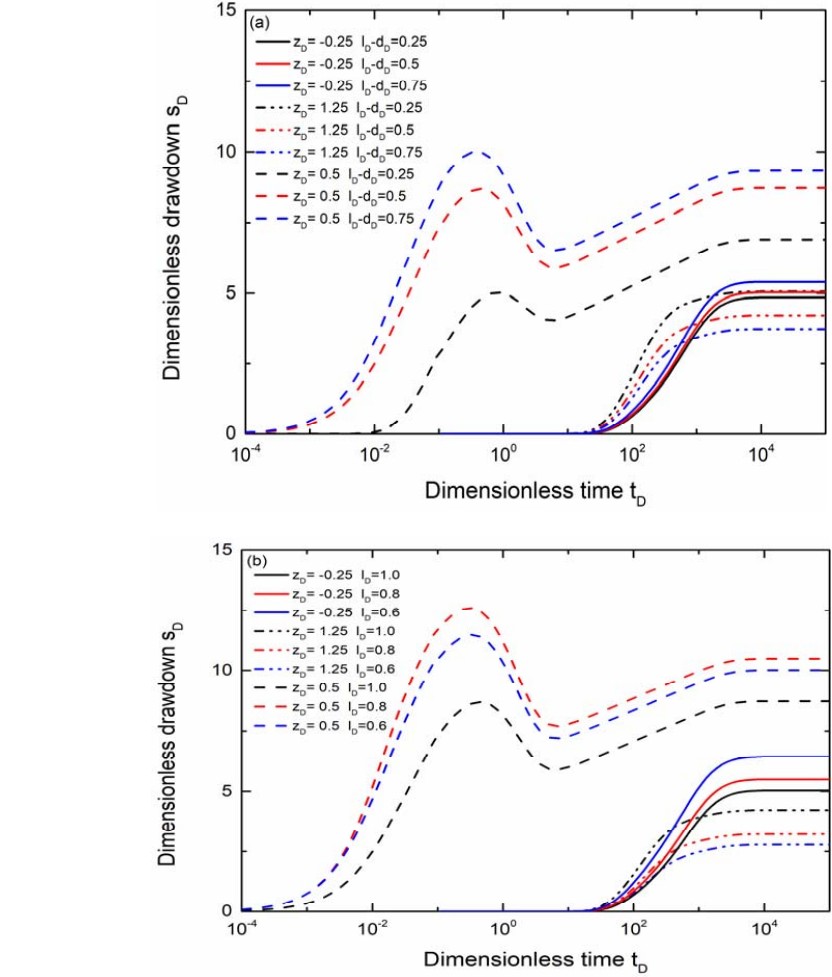



Fig. 10 Drawdown responses in the pumped layer and unpumped layers (Case 3) with $r_D =$
0.1, $\kappa_1 = \kappa_2 = 10^{-2}$, $\alpha_{Dz2} = \alpha_{Dr2} = 2 \times 10^{-4}$, $\alpha_{Dz3} = \alpha_{Dr3} = 2 \times 10^{-4}$, $\alpha_D = 0.8$, $Q_{1D} = 2.5$, $B_{D2} = 1.5$,
$B_{D3} = 0.5$ (a) for different well screen length, in which $l_D = 1.0$ (b) for various depth of well





screen within the middle pumped layer, where $l_D$ - $d_D$ = 0.5.

One of the main contributions in this study is that the established general analytical

model considered the effect of the well partial penetration, Fig. 10 shows the drawdown
changes for Case 3 ($r_D$ = 0.1) in the middle-pumped layer ($z_D$ = 0.5) and unpumped layers ($z_D$
= 1.25 and -0.25). Especially, Fig. 10 (a) is for various well screen length and $l_D$ = 1.0, and
Fig. 10 (b) is for different vertical position of well screen within the middle-pumped layer
and the well screen length is fixed ($l_D$ - $d_D$ = 0.5). It can be seen from Fig. 10 that the length
and position of well screen have remarkable effect on the drawdown for all three layers. A
larger well screen length means that the middle drawdown of pumped layer is closer to the
position of well screen and the stored water is much easier to be released, resulting in a larger
drawdown of pumped layer, similarly, a smaller drawdown for the upper layer and a greater
drawdown for the lower unpumped layer can be seen in Fig. 10 (a) for Case 3. Additionally,
one can conclude from the above analysis shown in Fig. 5 (b) that the closer to the center of
the pumped well, the larger drawdown can be seen for all three layers, and the drawdown for
the lower layer is relatively larger than the late-time drawdown for the upper layer at the
same distance measured from the interface between the pumped layer and unpumped layer
for Case 3. The center point of the well screen for three different $l_D$ = 1.0, 0.8 and 0.6 is
respectively at $z_D$ = 0.75, 0.55 and 0.35, respectively. Thus, the pumped layer drawdown ($z_D$
= 0. 5) with $l_D$ = 0.6 is larger than that with $l_D$ = 1.0 and smaller than that with $l_D$ = 0.8, in the
same way, the upper unpumped layer drawdown ($z_D$ = 1.25) with $l_D$ = 0.8 is larger than that
with $l_D$ = 0.6 and smaller than that with $l_D$ = 1.0, and the lower unpumped layer drawdown
($z_D$ = -0.25) with $l_D$ = 0.8 is larger than that with $l_D$ = 1.0 and smaller than that with $l_D$ = 0.6.
Besides that, whatever the pumping well is located at the pumped layer, the pumping induced
drawdown in the lower unpumped layer is larger than that in the upper layer for Case 3.
**4. Discussion**
Based upon the presented solution, firstly, one can perform quantitative evaluation of the
dimensionless drawdown at any points within the general three-layer aquifer system with a
partial penetration pumping well in the middle layer. It is worth emphasizing again that the
developed solution not only has no any restrictions on the values of the thickness, hydraulic
conductivity, and specific storage for all three layers, but that for the length and location of
the well screen fixed in the pumped layer, thus, the generality of the obtained solution is the
main contribution of this study. Secondly, it is convenient to explore the influences of
variable discharge of pumping, aquifer thickness, anisotropy, well partial penetration, and the
type of top and bottom boundary on the groundwater flow problems in the aquifer system of
three-layer. Besides that, the present solutions have a powerful potentiality within
geotechnical engineering, petroleum engineering and groundwater resource development.
Another important application of the proposed solution is to identify the hydraulic parameters
of each layer with adopting the method of parameter estimation in conjunction with field
data.
Because the responses for a special case of aquitard-aquifer-aquifer system is mainly
explored for comparison with existing solutions, some suggestions can be obtained for using
the developed solutions in such a three-layer aquifer from the above analysis herein. First of
all, the well structure (screen position and length) in the pumped layer and the thickness of all





layers should be clearly determined. Secondly, the type of boundary at the top and bottom of
the aquifer system should be clarified with the use of the observed data of late-time
drawdown for parameter estimation. Thirdly, the feature of inflection point for the curve of
drawdown against time due to the effect of variable discharge can be used to estimate the
pumped layer parameters, and in such a case the *in situ* data of drawdown in vicinity of the
pumping well need to be collected. Fourthly, the data of early-time drawdown for unpumped
layers are suggested to determine their specific storage respectively, the datum of late-time
drawdown for unpumped layers can be applied to estimate their values of hydraulic
conductivities respectively.

However, a few limitations of this study are also need to be addressed. Firstly, the effects

of finite radius and wellbore storage on flow cannot be investigated in this study because of
the assumption of infinitesimal radius of the pumping well. Secondly, the three-dimensional
transient responses in three-layer aquifer system have not been discussed with the condition
of constant-drawdown pumping, other type of variable-rate pumping (e.g. sinusoidal
pumping, piecewise-linear pumping), etc. Thirdly, the heterogeneity of the aquifer and
varying/non-uniform thickness of each layer are not taken into consideration. Fourthly, the
slope of each layer and the influence of finite or non-uniform well skin are not considered as
well. Fifthly, the effect of a finite or irregular lateral boundary is not analyzed. The
investigation for these subjects is much needed in details in the future.
**5. Summary and conclusions**

A general semi-analytical dimensionless drawdown solution in an anisotropic aquifer

system of three-layer caused by a partial penetration well pumped at a variable discharge is
developed by means of Laplace-Hankel transformation taking account of the interface flow.
Most importantly, three widely used types of boundary conditions at the top and bottom are
considered that include a zero-drawdown boundary for Case1 or a no-flow boundary for Case
2, and a constant-head boundary at the top in combination with a no-flux boundary at the
bottom for Case 3. The time-domain solutions are evaluated by performing numerical
inversion of the transformations of Laplace and Hankel. The present solutions encompass
some previously known solutions caused by a full or partial penetration pumping well in an
aquifer system of two-layer or single-layer as subsets. The three-dimensional transient
drawdown in the entire aquifer system pumped by a partial penetration well having a
discharge with exponentially decaying function in the middle layer is explored as an example
of illustration. From this study, one can conclude the following main findings:

(1) The pumped layer drawdown for Hantush (1960) with neglecting vertical flow in the

pumped layer and horizontal flow in the unpumped layer and the use of the Hantush-Jacob
approximation is greater that of this work for Case 2, especially at the early pumping time for
a fully penetrating well, and the unpumped layers drawdown for Hantush (1960) are greater
than that for present study.

(2) The effect of variable discharge describing an exponential decline function of

pumping time mainly affects the drawdown of the pumped layer, and a noticeable feature of
inflection points can be seen at the stage of the decay of well discharge and the region nearby
the well of pumping.

(3) The type of boundary at the top and bottom of the aquifer system has no influence on

the early- and intermediate-drawdown, but the drawdown at late pumping time for Case 3 is





greater than that for Case 1 and smaller than that for Case 2 in all three layers.
(4) A smaller anisotropy ratio (meaning a smaller vertical/horizontal permeability ratio)
of the pumped layer results in a larger pumped layer drawdown and a smaller unpumped
layer drawdown over the whole pumping times. The anisotropy of the unpumped layers ($K_D$)
mainly affects the drawdown in the unpumped layer and a larger anisotropy ratio ($K_D$) leads
to a larger drawdown of unpumped layer.
(5) The anisotropy of the unpumped layers significantly affects the drawdown in the
aquifer system without large contrast of hydraulic conductivity between the unpumped layers
and the pumped layer during entire pumping period.
(6) The drawdown nearby the pumping well in all three layers are significantly affected
by the length and position of well screen in the pumped layer at the entire time, and a larger
drawdown can be seen at the position of a smaller distance to the midpoint of the well screen.
**Author contributions.** F.QG., and F.XL., conceived the presented idea, F.QG., developed the
solutions and codes for the model, F.QG., and Z.HB., performed the results and discussion.
F.XL., and Z.HB., supervised the findings of the study. All authors contributed to the writing
and the final paper.
**Competing interest.** The authors declare that they have no conflict of interest.
**Acknowledgements.** This research was partially funded by the National Natural Science
Foundation of China (No. 41702336) and the research project for Wuhan Municipal
Construction Group Co., Ltd. (No. wszky201820).




**Appendix A. Derivations of solutions for different cases**


The Laplace and Hankel transformation technique are sequentially applied to Eqs. (17) –
(33), one can obtain the following Laplace-Hankel domain governing equations of flow in the
middle-pumped aquifer
$$\frac{\partial^2 \hat{\bar{s}}_{D1}}{\partial z_D^2} - \xi_1 \hat{\bar{s}}_{D1} = \frac{1}{\alpha_{Dz1}} \lim_{r_D \to 0} r_D \frac{\partial^2 \bar{s}_{D1}}{\partial r_D} \quad (A1)$$
with
$$\lim_{r \to 0} r_D \frac{\partial \bar{s}_{D1}}{\partial r_D} = \begin{cases} 0 & l_D < z_D \leq 1 \\ -\dfrac{2\bar{Q}(p)}{l_D - d_D} & d_D \leq z_D \leq l_D \\ 0 & 0 \leq z_D < d_D \end{cases} \quad (A2)$$
and the variable discharge used in this study is expressed in Eq. (5), one can obtain,
$$\bar{Q}(p) = \frac{1}{p} + \frac{Q_{1D} - 1}{p + \alpha_D} \quad (A3)$$
Substituting Eq. (A3) into Eq. (A2) results in
$$\lim_{r_D \to 0} r_D \frac{\partial \bar{s}_{D1}}{\partial r_D} = \begin{cases} 0 & l_D < z_D \leq 1 \\ -\dfrac{2}{l_D - d_D}\left(\dfrac{1}{p} + \dfrac{Q_{1D} - 1}{p + \alpha_D}\right) & d_D \leq z_D \leq l_D \\ 0 & 0 \leq z_D < d_D \end{cases} \quad (A4)$$
To derive the solution of Eq. (A1), using the method proposed by Neuman (1974), the
dimensionless drawdown for the middle-pumped layer ( $s_{D1}$ ) can be divided into the following
form and written in Laplace-Hankel space as:
$$\hat{\bar{s}}_{D1} = \hat{\bar{u}}_D + \hat{\bar{v}}_D \quad (A5)$$
in which $\hat{\bar{u}}_D$ designates the Laplace-Hankel domain drawdown solution in a confined aquifer
caused by a partial penetration pumping well, and the final expression of $\hat{\bar{u}}_D$ written in Eq.
(33) can be obtained by complying with the analogous process adopted by Feng and Zhan
(2019). $\hat{\bar{v}}_D$ satisfies Eqs. (17) and (24)-(27).



Under this circumstance, the governing equation of $\hat{\bar{v}}_D$ becomes

$\dfrac{\partial^2 \hat{\bar{v}}_D(\lambda, z_D, p)}{\partial z_D^2} - \xi_1^2 \hat{\bar{v}}_D(\lambda, z_D, p) = 0$    (A6)

By analogy, the governing equations of the upper and lower unpumped layer are

respectively rewritten as
$\dfrac{\partial^2 \hat{\bar{s}}_{D2}(\lambda, z_D, p)}{\partial z_D^2} - \xi_2^2 \hat{\bar{s}}_{D2}(\lambda, z_D, p) = 0$    (A7)
and
$\dfrac{\partial^2 \hat{\bar{s}}_{D3}(\lambda, z_D, p)}{\partial z_D^2} - \xi_3^2 \hat{\bar{s}}_{D3}(\lambda, z_D, p) = 0$    (A8)

The interface boundary conditions at $z_D = 1$ given in Eqs. (24) and (25) become

$\hat{\bar{u}}_D(\lambda, 1, p) + \hat{\bar{v}}_D(\lambda, 1, p) = \hat{\bar{s}}_{D2}(\lambda, 1, p),$    $z_D = 1$    (A10)
$\dfrac{\partial \hat{\bar{v}}_D(\lambda, z_D, p)}{\partial z_D} = \kappa_1 \dfrac{\partial \hat{\bar{s}}_{D2}(\lambda, z_D, p)}{\partial z_D},$    $z_D = 1$    (A11)

And considering the boundary conditions at $z_D = 0$ expressed in Eqs. (26) and (27), one

can obtain
$\hat{\bar{u}}_D(\lambda, z_D, p) + \hat{\bar{v}}_D(\lambda, z_D, p) = \hat{\bar{s}}_{D3}(\lambda, z_D, p),$    $z_D = 0$    (A12)
$\dfrac{\partial \hat{\bar{v}}_D(r_D, z_D, p)}{\partial z_D} = \kappa_2 \dfrac{\partial \hat{\bar{s}}_{D3}(r_D, z_D, p)}{\partial z_D},$    $z_D = 0$    (A13)

Finally, the top and bottom boundary conditions given in Eqs. (28)-(33) can be rewritten

as:

For Case 1,

$\hat{\bar{s}}_{D2}(r_D, z_D, p) = 0,$   $z_D = B_{D2}$    (A14)
$\hat{\bar{s}}_{D3}(r_D, z_D, p) = 0,$   $z = -B_{D3}$    (A15)

For Case 2,



$\quad \dfrac{\partial \hat{\bar{s}}_{D2}\left(r_D, z_D, p\right)}{\partial z} = 0,\ \ z_D = B_{D2}$ (A16)
$\quad \dfrac{\partial \hat{\bar{s}}_{D3}\left(r_D, z_D, p\right)}{\partial z_D} = 0,\ \ z_D = -B_{D3}$ (A17)
$\quad$ and
$\quad\quad$ for Case 3,
$\quad \hat{\bar{s}}_{D2}\left(r_D, z_D, p\right) = 0,\ \ z_D = B_{D2}$ (A18)
$\quad \dfrac{\partial \hat{\bar{s}}_{D3}\left(r_D, z_D, p\right)}{\partial z_D} = 0,\ \ z_D = -B_{D3}$ (A19)
$\quad\quad$ The general solution for Eq. (A6) is
$\quad \hat{\bar{v}}_D\left(\lambda, z_D, p\right) = c_1 e^{\xi_1 z_D} + c_2 e^{-\xi_1 z_D}$ (A20)
$\quad\quad$ Substituting Eq. (A20) into Eq. (A5), one can write
$\quad \hat{\bar{s}}_{D1} = \hat{\bar{u}}_D\left(\lambda, z_D, p\right) + c_1 e^{\xi_1 z_D} + c_2 e^{-\xi_1 z_D}$ (A21)
$\quad\quad$ The general solutions of Eqs. (A7) and (A8) for flow in the upper and lower unpumped
layers can be expressed, respectively, as
$\quad \hat{\bar{s}}_{D2} = c_3 e^{\xi_2 z_D} + c_4 e^{-\xi_2 z_D}$ (A22)
and
$\quad \hat{\bar{s}}_{D3} = c_5 e^{\xi_3 z_D} + c_6 e^{-\xi_3 z_D}$ (A23)
$\quad\quad$ Using the continuity boundary conditions of Eqs. (A10)-(A13) leads to
$\quad \hat{\bar{u}}_D\left(\lambda, 1, p\right) + c_1 e^{\xi_1} + c_2 e^{-\xi_1} - c_3 e^{\xi_2} - c_4 e^{-\xi_2} = 0$ (A24)
$\quad c_1 e^{\xi_1} - c_2 e^{-\xi_1} - \gamma_1\left(c_3 e^{\xi_2} - c_4 e^{-\xi_2}\right) = 0$ (A25)
$\quad \hat{\bar{u}}_D\left(\lambda, 0, p\right) + c_1 + c_2 - c_5 - c_6 = 0$ (A26)
and
$\quad c_1 - c_2 - \gamma_2\left(c_5 - c_6\right) = 0$ (A27)





Applying the top and bottom boundary conditions Eqs. (A10)-(A13), one can write
Case 1,
$c_3 e^{\xi_2 B_{D2}} + c_4 e^{-\xi_2 B_{D2}} = 0$  (A28)
$c_5 e^{-\xi_3 B_{D3}} + c_6 e^{\xi_3 B_{D3}} = 0$  (A29)
Case 2,
$c_3 e^{\xi_2 B_{D2}} - c_4 e^{-\xi_2 B_{D2}} = 0$  (A30)
$c_5 e^{-\xi_3 B_{D3}} - c_6 e^{\xi_3 B_{D3}} = 0$  (A31)
and
Case 3,
$c_3 e^{\xi_2 B_{D2}} + c_4 e^{-\xi_2 B_{D2}} = 0$  (A32)
$c_5 e^{-\xi_3 B_{D3}} - c_6 e^{\xi_3 B_{D3}} = 0$  (A33)
Solving equations consisting of expressions (A24)–(A27) and (A28)–(A29), the
coefficients that need to be determined for Case 1 are
$c_1 = \dfrac{2}{\chi_1} \left\{ \begin{array}{l} \hat{\bar{u}}(r_D, 0, p) e^{-\xi_1} \gamma_2 \left[ (\cosh\theta_1 + \cosh\theta_2)\gamma_1 - (\sinh\theta_1 + \sinh\theta_2) \right] \\ -\hat{\bar{u}}(r_D, 1, p) \gamma_1 \left[ (\cosh\theta_1 + \cosh\theta_2)\gamma_2 + \sinh\theta_1 - \sinh\theta_2 \right] \end{array} \right\}$  (A34a)
and
$c_2 = -\dfrac{2}{\chi_1} \left\{ \begin{array}{l} 2\hat{\bar{u}}(r_D, 0, p) e^{\xi_1} \gamma_2 \left[ (\cosh\theta_1 + \cosh\theta_2)\gamma_1 + \sinh\theta_1 + \sinh\theta_2 \right] \\ -2\hat{\bar{u}}(r_D, 1, p) \gamma_1 \left[ (\cosh\theta_1 + \cosh\theta_2)\gamma_2 - (\sinh\theta_1 - \sinh\theta_2) \right] \end{array} \right\}$  (A34b)
with $c_3$, $c_4$, $c_5$, and $c_6$ written by $c_1$ and $c_2$.
$c_3 = \dfrac{1}{2\gamma_1} e^{-\xi_2} \left[ c_1 e^{\xi_1}(\gamma_1 + 1) + c_2 e^{-\xi_1}(\gamma_1 - 1) + \gamma_1 \hat{\bar{u}}_D(r_D, 1, p) \right]$  (A34c)
$c_4 = \dfrac{1}{2\gamma_1} e^{-\xi_2} \left[ c_1 e^{\xi_1}(\gamma_1 - 1) + c_2 e^{-\xi_1}(\gamma_1 + 1) + \gamma_1 \hat{\bar{u}}_D(r_D, 1, p) \right]$  (A34d)
$c_5 = \dfrac{1}{2\gamma_2} \left[ c_1(\gamma_2 + 1) + c_2(\gamma_2 - 1) + \gamma_2 \hat{\bar{u}}_D(r_D, 0, p) \right]$  (A34e)
$c_6 = \dfrac{1}{2\gamma_2} \left[ c_1(\gamma_2 - 1) + c_2(\gamma_2 + 1) + \gamma_2 \hat{\bar{u}}_D(r_D, 0, p) \right]$  (A34f)
where





$$\chi_1 = 2(1+\gamma_1)(1+\gamma_2)\sinh(\xi_1+\theta_1) + 2(1-\gamma_1)(1-\gamma_2)\sinh(\xi_1-\theta_1)$$
$$-2(1+\gamma_1)(1-\gamma_2)\sinh(\xi_1+\theta_2) - 2(1-\gamma_1)(1+\gamma_2)\sinh(\xi_1-\theta_2)$$
(A34g)

Similarly, solving equations including Eqs. (A20)–(A24) and Eqs. (A28)–(A29), the
related coefficients used in Case 2 yield
$$c_1 = \frac{2}{\chi_2}\left\{\begin{array}{l}\hat{\bar{u}}(r_D,0,p)e^{-\xi_1}\gamma_2\left[(\cosh\theta_2-\cosh\theta_1)\gamma_1+(\sinh\theta_1-\sinh\theta_2)\right]\\ +\hat{\bar{u}}(r_D,1,p)\gamma_1\left[(\cosh\theta_1-\cosh\theta_2)\gamma_2+(\sinh\theta_1+\sinh\theta_2)\right]\end{array}\right\}$$
(A35a)

$$c_2 = -\frac{2}{\chi_2}\left\{\begin{array}{l}\hat{\bar{u}}(r_D,0,p)e^{\xi_1}\gamma_2\left[(\cosh\theta_2-\cosh\theta_1)\gamma_1-(\sinh\theta_1-\sinh\theta_2)\right]\\ +\hat{\bar{u}}(r_D,1,p)\gamma_1\left[(\cosh\theta_1-\cosh\theta_2)\gamma_2-(\sinh\theta_1+\sinh\theta_2)\right]\end{array}\right\}$$
(A35b)

$$c_3 = \frac{1}{2\gamma_1}e^{-\xi_2}\left[c_1 e^{\xi_1}(\gamma_1+1)+c_2 e^{-\xi_1}(\gamma_1-1)+\gamma_1\hat{\bar{u}}_D(r_D,1,p)\right]$$
(A35c)

$$c_4 = \frac{1}{2\gamma_1}e^{-\xi_2}\left[c_1 e^{\xi_1}(\gamma_1-1)+c_2 e^{-\xi_1}(\gamma_1+1)+\gamma_1\hat{\bar{u}}_D(r_D,1,p)\right]$$
(A35d)

$$c_5 = \frac{1}{2\gamma_2}\left[c_1(\gamma_2+1)+c_2(\gamma_2-1)+\gamma_2\hat{\bar{u}}_D(r_D,0,p)\right]$$
(A35e)

$$c_6 = \frac{1}{2\gamma_2}\left[c_1(\gamma_2-1)+c_2(\gamma_2+1)+\gamma_2\hat{\bar{u}}_D(r_D,0,p)\right]$$
(A35f)

$$\chi_2 = -2(1+\gamma_1)(1+\gamma_2)\sinh(\xi_1+\theta_1) - 2(1-\gamma_1)(1-\gamma_2)\sinh(\xi_1-\theta_1)$$
$$-2(1+\gamma_1)(1-\gamma_2)\sinh(\xi_1+\theta_2) - 2(1-\gamma_1)(1+\gamma_2)\sinh(\xi_1-\theta_2)$$
(A35g)

In the same way, one can solve the equations using Eqs. (A20)–(A24) and (A27), the
results for Case 3 are
$$c_1 = \frac{2}{\chi_3}\left\{\begin{array}{l}\hat{\bar{u}}(r_D,0,p)e^{-\xi_1}\gamma_2\left[(\sinh\theta_2-\sinh\theta_1)\gamma_1+(\cosh\theta_1-\cosh\theta_2)\right]\\ +\hat{\bar{u}}(r_D,1,p)\gamma_1\left[(\sinh\theta_1-\sinh\theta_2)\gamma_2+(\cosh\theta_1+\cosh\theta_2)\right]\end{array}\right\}$$
(A36a)

$$c_2 = -\frac{2}{\chi_3}\left\{\begin{array}{l}2\hat{\bar{u}}(r_D,0,p)e^{\xi_1}\gamma_2\left[(\sinh\theta_2-\sinh\theta_1)\gamma_1-(\cosh\theta_1-\cosh\theta_2)\right]\\ +2\hat{\bar{u}}(r_D,1,p)\gamma_1\left[(\sinh\theta_1-\sinh\theta_2)\gamma_2-(\cosh\theta_1+\cosh\theta_2)\right]\end{array}\right\}$$
(A36b)

$$c_3 = \frac{1}{2\gamma_1}e^{-\xi_2}\left[c_1 e^{\xi_1}(\gamma_1+1)+c_2 e^{-\xi_1}(\gamma_1-1)+\gamma_1\hat{\bar{u}}_D(r_D,1,p)\right]$$
(A36c)

$$c_4 = \frac{1}{2\gamma_1}e^{-\xi_2}\left[c_1 e^{\xi_1}(\gamma_1-1)+c_2 e^{-\xi_1}(\gamma_1+1)+\gamma_1\hat{\bar{u}}_D(r_D,1,p)\right]$$
(A36d)

$$c_5 = \frac{1}{2\gamma_2}\left[c_1(\gamma_2+1)+c_2(\gamma_2-1)+\gamma_2\hat{\bar{u}}_D(r_D,0,p)\right]$$
(A36e)

$$c_6 = \frac{1}{2\gamma_2}\left[c_1(\gamma_2-1)+c_2(\gamma_2+1)+\gamma_2\hat{\bar{u}}_D(r_D,0,p)\right]$$
(A36f)

$$\chi_3 = -2(1+\gamma_1)(1+\gamma_2)\cosh(\xi_1+\theta_1) + 2(1-\gamma_1)(1-\gamma_2)\cosh(\xi_1-\theta_1)$$
$$-2(1+\gamma_1)(1-\gamma_2)\cosh(\xi_1+\theta_2) + 2(1-\gamma_1)(1+\gamma_2)\cosh(\xi_1-\theta_2)$$
(A36g)



Finally, substituting the obtained coefficients for various cases above into Eq. (A21) –
Eq. (A23) respectively, and performing inverse Hankel transform can be, after some
mathematical manipulation details, written in Eqs. (29) – (37). So far, semi-analytical
solutions in the pumped and unpumped layers are derived.

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
