# Peer review of "Three-dimensional transient flow to a partially penetrated well with variable discharge in a general three-layer aquifer system"

_Hydrology and Earth System Sciences, 2020_

## Referee Comment (RC1) · Shlomo P. Neuman (Referee) · 7 Jan 2021

By Shlomo P. Neuman

This manuscript provides a mathematical solution to Hankel and Laplace transformed equations describing flow to a line sink of variable strength partially penetrating a hydraulically anisotropic aquifer of finite thickness, confined above and below by anisotropic aquitards of finite thickness, all three layers extending horizontally to infinity. Horizontal boundaries at the top and bottom of this three-layer system are assigned prescribed head (Case 1), zero flux (Case 2), or prescribed head at top and zero flux at bottom (Case 3). The solutions are transformed back into real space-time coordinates numerically.

Although the authors consider their model to represent a general three-layer aquifer system, there are two indications that the top and bottom layers must be aquitards: 1) Pumping takes place only from the middle layer, suggesting that the top and bottom layers are not productive, and 2) top and bottom boundary conditions are those of the classical Hantush-Jacob "leaky aquifer" model, which the authors consistently compare with theirs. This comparison leads the authors to conclude that theirs is a more general model because it allows for partial penetration, anisotropy, multidirectional flow and variable pumping rate.

In reality, the proposed solution is severely limited by the replacement of aquifers above and below the two aquitards with artificially imposed boundary conditions and by treating the top and bottom layers as aquitards rather than, potentially, productive aquifer layers. The authors forget to mention that analytical solutions exist for more realistic multiaquifer systems with aquifers above and below their aquitards (replaced in their model by artificial Hantush-Jacob boundary conditions); see Neuman (1968), Neuman and Witherspoon (1969) and Li and Neuman (2007). Though it is true that these models restrict flow in aquifers to horizontal, flow in aquitards to vertical, and flow to a fully penetrating well, those restrictions have been demonstrated by Neuman (1968) and Neuman and Witherspoon (1969) to be much less severe than implied by the authors.

The authors likewise forget that any analytical expression for constant pumping rate is easily generalized to the case of variable pumping rate through the temporal superposition of elementary expressions; this has been done routinely for years in software packages such as Aqtesolv.

In summary, I find the manuscript to be somewhat misleading in its claim of providing a useful general solution to well flow in a multilayer aquifer system, and the proposed solution to be at best of marginal interest to hydrologists.

---

## Referee Comment (RC2) · Ed J.M. Veling (Referee) · 14 Jan 2021

**Referee Report on**
**"Three-dimensional transient flow to a partially penetrated well with variable discharge in a general three-layer aquifer system" by Qinggao Feng, Xiaola Feng, and Hongbin Zhan**

offered for publication to *Hydrology and Earth System Sciences* (HESS), MS No.: hess-2020-58

E.J.M. Veling

14th January 2021

**1    General comments**

This paper treats the case of three hydraulically anisotropic aquifer layers of finite thickness where in the middle layer a partially penetrating well is active. There are no restrictions with respect to the aquifer parameters $K_{ri}$ ($= K_{hi}$), $K_{zi}$ ($= K_{vi}$) and $S_{si}$. In fact, this paper has the same set-up as the paper by Feng *et al.* (2020) in which the well is active in the lower layer or to the paper by Feng *et al.* (2019) for a two layer case in which the well is active in the lower layer. These three papers are based on the assumption made by Neuman (1974) to handle the well by a solution of Hantush (1964) for drawdown in a confined aquifer due to pumping from a well that partially penetrates the aquifer. This assumption is not mentioned in the introduction where the suggestion has been raised that the followed approach is exact, which is not. This simplifying assumption is somewhat hidden in the Appendix A of this paper, namely in requirements (A11) and (A13) where the terms $\partial \widehat{\widehat{u}}_D(\lambda, 1, p)/\partial z_D$, resp. $\partial \widehat{\widehat{u}}_D(\lambda, 0, p)/\partial z_D$ are missing.

The main advantage of this approach however is that the authors of this paper and of the papers Feng (2019, 2020) (and for example also the paper by Malama *et al.* (2008)) end up with a Laplace-Hankel transform consisting of an integral with a closed form expression for the integrand. Of course, for every Laplace variable $p$ the Hankel integral has to be repeated because the parameters in the integrand depend on $p$, but the integrand is a single expression, albeit complicated.

It is remarkable that the authors do not mention the paper by Veling & Maas (2009) [= VeMa] where there is not such a restriction described above with respect to the conditions of the flux at the interface of the layers. Moreover, in the VeMa-paper the solution has been given for an arbitrary number of layers with much freedom with respect to the upper and lower boundary conditions (Dirichlet, Neumann, Robin boundary conditions) and allows for more that one well screen (injection and extraction in the same well bore, even in the same layer, *e.g.*). In VeMa three different strategies are described with respect to the order of transformations. The authors of the paper under review use strategy 5.3 "Integral transform in terms of $t$ and $r$" in VeMa,.but do not proceed to take into account the influence of the well into the upper and lower layer. In VeMa particle tracking has been applied for a 6-layer aquifer with 3 well screens in the same well bore as an example, among others. In Feng (2020) almost the same authors as the current ones mention VeMa where they only say that their time-dependent extraction function is a generalization with respect to VeMa (which can easily for accounted for in VeMa), but forget to say that VeMa is more general and exact with respect to the conditions along the interface of the layers.

The authors do not specify in which way they found the quite complicated expressions: by hand or by a formula manipulation package? If the authors have used some formula manipulation package, that should be stated clearly. The mathematical problem to be solved for the problem treated by the authors consists of solving three times (for the three Cases A, B and C) a set of 6 equations with 6 unknowns (Appendix A). The numerical approach sounds good with modern tools (de Hoog et al. (1982) and Ogata (2005)).

The authors apply their results in a consistent way for an isotropic system with a fully penetrating well and compare their results extensively with the existing litterature. The authors study also an anisotropic system with a full and a partially (half the thickness of the aquifer) penetrating well. All their results are good understandable and explained.

The overall writing is good and precise, but see below w.r.t. the References.

**2 Some remarks**

Line 786 and 790: The approximation described above should be mentioned.
Line 657: It is not clear what exactly is meant with the expression "that the middle drawdown of pumped layer is closer to the position of well screen".
Line 864: References. A number of referenced papers are not mentioned in the References.

**3 Some minor remarks**

Line 223: Table 1: The extra horizontal line is confusing. The variable $\alpha_{ri}$ and $\alpha_{zi}$ are non dimensionless.
Line 775: Eq. (33) should be Eq. (34).

**4 Final Remark**

Overall, the paper serves as an useful approximation for the specific problem at hand, but compare the remarks made above. Therefore, this reviewer judges that the claims in the paper (Line 324 and Line 679) should be somewhat more modest. It should be interesting to compare their solution to the general solution given by VeMa and to find out under which conditions the conclusions of the authors are still valid. This reviewer suggests that it occurs if the well screen is large compared to the thickness of the layer which implies that the partially penetrating well induces mainly a radial flow.

**5 References**

Qinggao Feng, Xiang Yuan, Hongbin Zhan, 2019,
"Flow to a partially penetrating well with variable discharge in an anisotropic two-layer aquifer system",
*Journal of Hydrology* 578, https://doi.org/10.1016/j.jhydrol.2019.124027

Qinggao Feng, Yu Luo, Hongbin Zhan, 2020,
"Three-dimensional response to a partially penetration well pumping in a general anisotropic three-layer aquifer system",
*Journal of Hydrology* 585, https://doi.org/10.1016/j.jhydrol.2020.124850

M.S. Hantush, 1964,
"Hydraulics of Wells",
in Advances in Hydroscience, vol. 1. Academic, New York, pp. 282–432

De Hoog, F.R., Knight, J.H., Stokes, A.N., 1982,
"An improved method for numerical inversion of Laplace transforms",
SIAM J. Sci. Stat. Comput. 3 (3), 357–366

Bwalya Malama, Kristopher L. Kuhlman, Warren Barrash, 2008,
"Semi-analytical solution for flow in a leaky unconfined aquifer toward a partially penetrating pumping well",
Journal of Hydrology 356:234– 244

Shlomo P. Neuman, 1974,
"Effect of Partial Penetration on Flow in Unconfined Aquifers Considering Delayed Gravity Response",
*Water Resources Research* 10:303-312

Ogata, H., 2005,
"A numerical integration formula based on the Bessel functions",
Publications of the Research Institute for Mathematical Sciences, 41(4), 949–970

E.J.M. Veling, C. Maas, 2009,
"Strategy for solving semi-analytically three-dimensional transient flow in a coupled N-layer aquifer system",
*Journal of Engineering Mathematics* 64:145–161, https://doi.org/10.1007/s10665-008-9256-9

---

## Referee Comment (RC3) · Anonymous Referee #3 · 17 Jan 2021

This is my review of the manuscript submitted by Feng, Feng and Zhan to HESS. The manuscript describes an analytical solution for a confined two-dimensional axisymmetric flow problem with three layers with variable discharge rate. The solution appears correct, but not particularly novel. Its difference from several other existing analytical solutions is a technicality (there are many layered analytical flow solutions in the literature).

The authors do not present any data or comparison against reality to justify the analytical solution design. It is easier to re-derive an analytical solution for a given problem than it is to drill a well. If the authors presented field data and used this solution to gain

insight into observed physical behavior for a real-world system, I think the description of this analytical solution could be relegated to an appendix of that paper.

Specific Comments

1) The authors call their solution "three dimensional," but it is only two-dimensional (r and z).

2) Lines 85-109: the authors build up a straw man about how difficult and inaccurate numerical solutions are, to lead into their discussion of how general and robust their analytical solution is. I definitely believe analytical solutions are useful and have their place, but they are not "better" than numerical models. It may be more appropriate to discuss how analytical solutions can be quick to evaluate (but very complex analytical solutions that are essentially "numerical" like this one are often not so quick to evaluate numerically), and therefore can be used in sensitivity analyses to gain insight into physical behavior through inverse modeling problems. Most of the comments about pitfalls related to numerical solution also apply to analytical solutions. The authors performed a double integral transform, and numerically invert both of these transforms. Numerically evaluating an analytical solution that involves two integral transforms can lead to more potentially dubious numerical manipulations than involved in most "numerical models."

2a) How many terms were used in the numerical inverse Laplace and Hankel transform algorithms? (what was the criteria used to ensure the solution had converged?)

2b) What criteria was use to chose the convergence of these series? Fixed number of terms? Was the solution compared with different numbers of terms?

2c) What order were the equations inverted (inverse Hankel first or inverse Laplace first)? Does the solution depend on the order they are inverted?

2d) Many of the terms in the analytical solution involve differences of exponentials or hyperbolic trigonometric functions. Subtraction of very large terms can lead to catastrophic cancellation, was this considered? Were the terms in the solution algebraically manipulated to minimize loss of significance? Could they be written in an equivalent manner that was more accurate than is written in the manuscript?

The authors simply point to Feng et al. (2020) and Liang et al. (2018) and do not discuss any details of the accuracy or convergence of their method for evaluating their "numerical" analytical solution (line 362). I would contend analytical solutions are more finicky and failure-prone than numerical models, so they require more careful scrutiny. The convergence of finite difference or finite element numerical models for solving confined groundwater flow (linear diffusion equations for a homogeneous problem) is pretty well-known and is not going to surprise anyone. Numerical models can also consider: 1) finite wellbore radius, 2) heterogeneity, 3) variable pumping rates, 4) non-linearities (e.g., the equation of state for water). I think the authors could cut down the section that discusses general "problems" with numerical models (lines 85-109) to a sentence. They could also cut down the section that talks about how general their analytical solution is. Both these un-needed sections could be replaced with discussion about the "numerical" details of evaluating their solution, which would actually be useful to someone who was going to try to implement this (the current general discussion about how much better analytical solutions are than numerical solutions in general is not useful).

3) The authors claim they have created a general and useful solution, but they also use an infinitessimal wellbore with no wellbore storage. Wellbore storage is very important, especially if you can have any range of aquifer properties in all three layers. All wells experience wellbore storage to some degree (unless it is a constant-head pumping test), the balance of the volume in the wellbore interval to the formation storage properties indicates whether or not it is significant. This solution will only be correct in the limiting case of small wellbore storage. The authors admit this (lines 636-645), but indicate that that will be coming in the next analytical solution.

4) Variability in the pumping rate is a trivial difference between this solution and other

solutions. Since the solutions is completely linear, Duhamel's theorem can be used to superimpose solutions that are pulses in time, or other combinations of steps on and off. The authors should provide some data or an example where this type of behavior (exponentially declining pumping rate) occurs. It is only included here because it is a simple case to consider in Laplace space, not because it is physically meaningful.

---

## Author Comment (AC1) · 27 Feb 2021

**Dear Professor Neuman:**

Upon the recommendation, we have carefully replied our manuscript HESS-2020-586 entitled "Three-dimensional transient flow to a partially penetrated well with variable discharge in a general three-layer aquifer system" after considering all your comments. The following is the point-by-point reply to all the comments.

**1.** This manuscript provides a mathematical solution to Hankel and Laplace transformed equations describing flow to a line sink of variable strength partially penetrating a hydraulically anisotropic aquifer of finite thickness, confined above and below by anisotropic aquitards of finite thickness, all three layers extending horizontally to infinity. Horizontal boundaries at the top and bottom this three-layer system are assigned prescribed head (Case 1), zero flux (Case 2), or prescribed head at top and zero flux at bottom (Case 3). The solutions are transformed back into real spacetime coordinates numerically. Although the authors consider their model to represent a general three-layer aquifer system, there are two indications that the top and bottom layers must be aquitards: 1) Pumping takes place only from the middle layer, suggesting that the top and bottom layers are not productive, and 2) top and bottom boundary conditions are those of the classical Hantush-Jacob "leaky aquifer" model, which the authors consistently compare with theirs. This comparison leads the authors to conclude that theirs is a more general model because it allows for partial penetration, anisotropy, multidirectional flow and variable pumping rate.

*Reply: Thank Dr. Neuman for his comments. In fact, Dr. Neuman and his co-authors have made tremendous contributions for developing the well flow theories since 1960 and our work here is inspired by their works and it represents our effort of pushing the work one step further to advance the analytical theories of well flow by relaxing some of the strict assumptions employed in present theoretical framework. Our responses are as follows.*

*Firstly, pumping taking place from the middle layer is simply an example of illustration of the methodology. As a matter of fact, the pumping can take place in either layer, and the mathematical modeling can be applied in a similar fashion based on the procedures documented in this investigation.*

*Secondly, pumping taking place from the middle layer does not necessarily mean that the top and bottom layers are aquitards. For instance, it could be that the middle layer is more productive than the upper and lower layers whose permeabilities are less than (but not several orders of magnitude smaller than) the permeability of the middle layer. For the upper and lower layers to be classified as aquitards, their permeabilities must be at least two, three, or even more orders of magnitude smaller than their counterpart of the middle layer. This study, however, does not impose such a strict constrain on the contrasts of permeabilities of different layers. Furthermore, it is entirely possible that the well may be installed in the middle layer (as a partially penetrating well) for a variety of reasons even though the upper and lower layers are also permeable and productive.*

*Thirdly, we agree with Dr. Neuman that the selection of the top and bottom boundary conditions follows the same line as what has been done by the traditional Hantush-Jacob "leaky aquifer" model. However, such a selection does not imply that the upper and lower layers must be*

*aquitards. For example, if there is an open surface body above the upper layer, the top boundary of the upper layer can be described using a prescribed-head or the first-kind of boundary condition, or if there is an extremely permeable aquifer above the less permeable (but definitely not an aquitard) upper layer and the permeability contrasts are more than two or three orders of magnitude, sometimes we may also approximate the top boundary of the upper layer as a prescribed-head boundary, provided that any flux exchange between the upper layer and the extremely permeable aquifer above the upper layer will not affect the hydraulic head in that extremely permeable aquifer noticeably. As another example, if a horizontal fracture exists above the upper layer or below the lower layer, and such a horizontal fracture extends sufficiently far from the domain of interest and is connected with a surface water body, the upper or lower boundary may also be described as a prescribed head boundary. As a third example, if an intact impermeable rock is underneath the lower layer, then the bottom boundary of the lower layer can be described using a prescribed zero flux boundary condition.*

*In summary, the choice of the top and bottom boundary condition does not necessarily imply that the upper and lower layer must be aquitards. One reason of using the classical Hantush-Jacob "leaky aquifer" types of boundary conditions here is for the sake of model comparison, because our model should be degenerated to the classical Hantush-Jacob "leaky aquifer" model when the upper and lower layers are indeed aquitards.*

*We do agree with the comment that we cannot simply limit to the classical Hantush-Jacob "leaky aquifer" types of boundary conditions. Thus, in the future, we need to investigate what other possible boundary conditions should be considered for a general three-layer aquifer system. We also feel that it is necessary to validate the model and the choices of boundary conditions using controlled laboratory experiments and field pumping tests as well in the future.*

**2.** In reality, the proposed solution is severely limited by the replacement of aquifers above and below the two aquitards with artificially imposed boundary conditions and by treating the top and bottom layers as aquitards rather than, potentially, productive aquifer layers. The authors forget to mention that analytical solutions exist for more realistic multiaquifer systems with aquifers above and below their aquitards (replaced in their model by artificial Hantush-Jacob boundary conditions); see Neuman (1968), Neuman and Witherspoon (1969) and Li and Neuman (2007). Though it is true that these models restrict flow in aquifers to horizontal, flow in aquitards to vertical, and flow to a fully penetrating well, those restrictions have been demonstrated by Neuman (1968) and Neuman and Witherspoon (1969) to be much less severe than implied by the authors.

***Reply:*** *The available models (Neuman,1968, Neuman and Witherspoon,1969 and Li and Neuman, 2007) restricted flow in aquifers to horizontal, flow in aquitards to vertical, and flow to a fully penetrating well. Apart from these, another important assumption used in those models is that mass exchange between two adjacent aquifers can be treated as a volumetric sink/source incorporated into the governing equations of flow in each individual layer, which cannot reflect the actual leakage process which occurs only at the interface of different layers and is better treated as an interface phenomenon as done in this investigation. This study has relaxed such restrictions or assumptions, and it represents the advancement of existing models to a physically based more general setting.*

**3.** The authors likewise forget that any analytical expression for constant pumping rate is easily generalized to the case of variable pumping rate through the temporal superposition of elementary expressions; this has been done routinely for years in software packages such as Aqtesolv.

*Reply: We acknowledge that one can obtain the analytical solution of variable pumping rate case by using the principle of superposition based on the solution of constant pumping rate case. But it is much needed to provide a simple and more direct (semi-)analytical solution (if possible) with variable discharge, as this is commonly encountered in real-world pumping tests according to our field experiences.*

**4.** In summary, I find the manuscript to be somewhat misleading in its claim of providing a useful general solution to well flow in a multilayer aquifer system, and the proposed solution to be at best of marginal interest to hydrologists.

*Reply: We have clearly clarified the applications and limitations of our solution in this study. The authors believe that this study is a valuable contribution to the subsurface hydrology by advancing the present well hydraulics theory one step further and it will be valuable to the hydrological community to keep pushing the boundary of the present stage of knowledge on the subject.*

On behalf of the authors
Sincerely Yours,
Hongbin Zhan

---

## Author Comment (AC2) · 27 Feb 2021

**Dear Professor Veling:**

Upon the recommendation, we have carefully replied our manuscript HESS-2020-586 entitled "Three-dimensional transient flow to a partially penetrated well with variable discharge in a general three-layer aquifer system" after considering all your comments. The following is the point-by-point reply to all the comments.

**1 General comments**

This paper treats the case of three hydraulically anisotropic aquifer layers of finite thickness where in the middle layer a partially penetrating well is active. There are no restrictions with respect to the aquifer parameters $K_{ri}$ (= $K_{hi}$), $K_{zi}$ (= $K_{vi}$) and $S_{si}$. In fact, this paper has the same set-up as the paper by Feng et al. (2020) in which the well is active in the lower layer or to the paper by Feng et al. (2019) for a two layer case in which the well is active in the lower layer. These three papers are based on the assumption made by Neuman (1974) to handle the well by a solution of Hantush (1964) for drawdown in a confined aquifer due to pumping from a well that partially penetrates the aquifer. This assumption is not mentioned in the introduction where the suggestion has been raised that the followed approach is exact, which is not. This simplifying assumption is somewhat hidden in the Appendix A of this paper, namely in requirements (A11) and (A13) where the terms $\partial \hat{\bar{u}}(\lambda, 1, p) / \partial z_D$ resp. $\partial \hat{\bar{u}}(\lambda, 0, p) / \partial z_D$ are missing.

***Reply:*** *Thank Dr. Veling for the detailed comments. When deriving the solution, we have used the same method adopted by Neuman (1974), as clarified in Appendix A. As suggested by Dr. Veling, we have revised the text in the introduction to address this concern.*

The main advantage of this approach however is that the authors of this paper and of the papers Feng (2019, 2020) (and for example also the paper by Malama et al. (2008)) end up with a Laplace-Hankel transform consisting of an integral with a closed form expression for the integrand. Of course, for every Laplace variable $p$ the Hankel integral has to be repeated because the parameters in the integrand depend on $p$, but the integrand is a single expression, albeit complicated.

It is remarkable that the authors do not mention the paper by Veling & Maas (2009) [= VeMa] where there is not such a restriction described above with respect to the conditions of the flux at the interface of the layers. Moreover, in the VeMa-paper the solution has been given for an arbitrary number of layers with much freedom with respect to the upper and lower boundary conditions (Dirichlet, Neumann, Robin boundary conditions) and allows for more that one well screen (injection and extraction in the same well bore, even in the same layer, e.g.). In VeMa three different strategies are described with respect to the order of transformations. The authors of the paper under review use strategy 5.3 "Integral transform in terms of $t$ and $r$" in VeMa, but do not proceed to take into account the influence of the well into the upper and lower layer. In VeMa particle tracking has been applied for a 6-layer aquifer with 3 well screens in the same wellbore as an example, among others. In Feng (2020) almost the same authors as the current ones mention VeMa where they only say that their time-dependent extraction function is a generalization with respect to VeMa (which can easily for accounted for in VeMa), but forget to

say that VeMa is more general and exact with respect to the conditions along the interface of the layers.

*Reply: We have revised the text to include a detailed analysis of the Veling & Maas (2009) [= VeMa] paper, and have also clarified the difference of the work of Veling & Maas (2009) and this study. The section 5.3 "Integral transform in terms of t and r" in VeMa has given the expression of hydraulic head (Eq. 40) using the Laplace transform to t, the Hankel transform to r and the generalized Fourier transform to z. The semi-analytical solution of VeMa did not formulate the closed-form expression for their solution in the z direction. In addition, one must calculate the eigenvalues $\lambda_m^2$ for every value of the Laplace transform parameters p and Hankel transform parameters α, and a matrix must be constructed to find the values of the eigenvalues of the eigenfunction. The solution became an simpler expression with the restrictive hypothesis of $K_{h,i}/S_{si} = \rho$. One can see that our present solutions does not apply any integral transform with respect to z in the process of the derivation, so it does not involve the problems of eigenvalue or eigenfunction. The section 5.3 "Integral transform in terms of t and r" in VeMa also provided another way used by Lenoach et al. (2004) to obtain the semi-analytical solution in a matrix form for a multi-layer aquifer system with no-flux boundary at the top and bottom (our solution for Case 2). However, the semi-analytical solutions with the other two types of boundary conditions (the prescribed head at top and bottom boundary (our solution for Case 1), or prescribed head at top and zero flux at bottom (our solution for Case 3)) used in our present study have not been considered by VeMa.*

The authors do not specify in which way they found the quite complicated expressions: by hand or by a formula manipulation package? If the authors have used some formula manipulation package, that should be stated clearly. The mathematical problem to be solved for the problem treated by the authors consists of solving three times (for the three Cases odnA, B and C) a set of 6 equations with 6 unknowns (Appendix A). The numerical approach sounds good with modern tools (de Hoog et al. (1982) and Ogata (2005)).

*Reply: We have revised the text to accommodate this suggestion. Briefly speaking, we have solved the 6 equations with 6 undetermined by using Maple.*

The authors apply their results in a consistent way for an isotropic system with a fully penetrating well and compare their results extensively with the existing litterature. The authors study also an anisotropic system with a full and a partially (half the thickness of the aquifer) penetrating well. All their results are good understandable and explained.

The overall writing is good and precise, but see below w.r.t. the References

*Reply: Thanks.*

**Some remarks**

Line 786 and 790: The approximation described above should be mentioned.

*Reply: We have revised the text in revised manuscript to address this concern.*

Line 657: It is not clear what exactly is meant with the expression "that the middle drawdown of

pumped layer is closer to the position of well screen".

***Reply:*** *It has been rewritten as "the drawdown at the middle part of pumped layer is closer to the position of well screen."*

Line 864: References. A number of referenced papers are not mentioned in the References.

***Reply:*** *We have carefully checked the Reference to ensure all papers are listed.*

**Some minor remarks**

Line 223: Table 1: The extra horizontal line is confusing. The variable $\alpha_{ri}$ and $\alpha_{zi}$ are non dimensionless.

***Reply:*** *Revised.*

Line 775: Eq. (33) should be Eq. (34).

***Reply:*** *Revised.*

**Final Remark**

Overall, the paper serves as a useful approximation for the specific problem at hand, but compare the remarks made above. Therefore, this reviewer judges that the claims in the paper (Line 324 and Line 679) should be somewhat more modest.

***Reply:*** *Revised.*

It should be interesting to compare their solution to the general solution given by VeMa and to find out under which conditions the conclusions of the authors are still valid.

***Reply:*** *The general solution shown in Eq. (40) is given by VeMa based on the application of the Laplace transform to t, the Hankel transform to r and the generalized Fourier transform to z. Our general solution is obtained by only using the Laplace and Hankel transform. The strategy for obtaining these two solutions has some similar features, but the final semi-analytical expression is different and the method to obtain the time-domain is also different. More importantly, we have verified our solution with comparison of available studies and numerical solution (Done by Feng et al.2020). We did not find similar comparison of analytical and numerical works in the paper of VeMa, which focused primarily on the "strategy" and "application" for particle tracking. In contrast, our study mostly aims to provide a relatively simple and direct general solution, to compare with the verified available studies, and to explore the drawdown behavior for the three-layer aquifer system induced by a variable discharge (exponentially declining pumping rate) with different top and bottom boundaries. After some careful considerations, we think the study of comparation presented in this paper is sufficient to support the conclusions and findings, so the comparison with the solution of VeMa is not conducted. However, it may be an interesting exercise to conduct a comparative investigation about different approaches, including this study, the study of VeMa, and other studies involving*

*multi-layer systems in the future to identify the advantages and disadvantages of those different approaches.*

This reviewer suggests that it occurs if the well screen is large compared to the thickness of the layer which implies that the partially penetrating well induces mainly a radial flow.

**Reply:** *Thanks for your suggestion. The content as suggested has been added in the revised manuscript.*

On behalf of the authors
Sincerely Yours,
Hongbin Zhan

References cited

Feng Q., Luo Y., Zhan H., 2020. Three-dimensional response to a partially penetration well pumping in a general anisotropic three-layer aquifer system, Journal of Hydrology, 2020, 585, 124850.

Lenoach, B., Ramakrishnan, T.S., Thambynayagam, R.K.M., (2004) Transient flow of a compressible fluid in a connected layered permeable medium. Trans Porous Media 57:153–169

E.J.M. Veling, C. Maas, 2009, Strategy for solving semi-analytically three-dimensional transient flow in a coupled N-layer aquifer system, Journal of Engineering Mathematics 64:145–161.

---

## Author Comment (AC3) · 27 Feb 2021

**Dear Referee 3:**

Upon the recommendation, we have carefully replied our manuscript HESS-2020-586 entitled "Three-dimensional transient flow to a partially penetrated well with variable discharge in a general three-layer aquifer system" after considering all your comments. The following is the point-by-point reply to all the comments.

This is my review of the manuscript submitted by Feng, Feng and Zhan to HESS. The manuscript describes an analytical solution for a confined two-dimensional axisymmetric flow problem with three layers with variable discharge rate. The solution appears correct, but not particularly novel. Its difference from several other existing analytical solutions is a technicality (there are many layered analytical flow solutions in the literature).
The authors do not present any data or comparison against reality to justify the analytical solution design. It is easier to re-derive an analytical solution for a given problem than it is to drill a well. If the authors presented field data and used this solution to gain insight into observed physical behavior for a real-world system, I think the description of this analytical solution could be relegated to an appendix of that paper.

*Reply: From the values of hydraulic parameters for each layer used in this study, one can see that the pumped or unpumped layer is composed of sandy soils or clay soil in nature, thus the observed physical behavior for a real-world system can be gained by using the new derived solution. We also feel that it is necessary to validate the model and the choices of boundary conditions using controlled laboratory experiments and field pumping tests as well in the future.*

Specific Comments

1) The authors call their solution "three dimensional," but it is only two-dimensional (*r* and *z*).

*Reply: We have corrected this in the revised manuscript.*

2) Lines 85-109: the authors build up a straw man about how difficult and inaccurate numerical solutions are, to lead into their discussion of how general and robust their analytical solution is. I definitely believe analytical solutions are useful and have their place, but they are not "better" than numerical models. It may be more appropriate to discuss how analytical solutions can be quick to evaluate (but very complex analytical solutions that are essentially "numerical" like this one are often not so quick to evaluate numerically), and therefore can be used in sensitivity analyses to gain insight into physical behavior through inverse modeling problems. Most of the comments about pitfalls related to numerical solution also apply to analytical solutions. The authors performed a double integral transform, and numerically invert both of these transforms. Numerically evaluating an analytical solution that involves two integral transforms can lead to more potentially dubious numerical manipulations than involved in most "numerical models."

*Reply: We have rewritten the content of L85-109 in the revised manuscript.*

2a) How many terms were used in the numerical inverse Laplace and Hankel transform algorithms? (what was the criteria used to ensure the solution had converged?)

*Reply: 40 terms of the series used in de Hoog algorithm has sufficient accuracy for the inverse solutions. For the inversion of Hankel transformation, the Ogata (2005) method has two free parameters, h, the step size, and N, the number of steps performed, which respectively determine the resolution and upper limit of the integration grid. These can be modified to accurately transform any function that theoretically converges. And we found that h=0.00001 and N=170 are enough for the inverse Hankel transformation in this study. We have clarified this issue in the revised manuscript.*

2b) What criteria was use to chose the convergence of these series? Fixed number of terms? Was the solution compared with different numbers of terms?

*Reply: For the inversion of Hankel transformation, the Ogata (2005) method has two free parameters, h, the step size, and N, the number of steps performed, which respectively determine the resolution and upper limit of the integration grid. These can be modified to accurately transform any function that theoretically converges. How to choose these values, and the estimated error of the transform under a given choice, are discussed in detail in the study of Ogata (2005) and the reader is referred to Ogata (2005) for more details. We have also clarified this issue in the revised manuscript.*

2c) What order were the equations inverted (inverse Hankel first or inverse Laplace first)? Does the solution depend on the order they are inverted?

*Reply: When we derive the general solution, the Laplace transform with respect to time t is applied, and then the Hankel transform with respect to r is carried out, after that we apply the inversion of Hankel transformation to obtain the semi-analytical solution in Laplace domain. One can obtain the final time-domain solution with application of inversion of Laplace transformation. For the inversion procedures, the readers can consult the detailed derivation shown in Appendix A in this study. We have further clarified the inversion order in the revised manuscript.*

2d) Many of the terms in the analytical solution involve differences of exponentials or hyperbolic trigonometric functions. Subtraction of very large terms can lead to catastrophic cancellation, was this considered? Were the terms in the solution algebraically manipulated to minimize loss of significance? Could they be written in an equivalent manner that was more accurate than is written in the manuscript?

*Reply: The method of Ogata (2005) shows good performance for similar exponentials or hyperbolic trigonometric functions (Liang et al. 2018; Feng et al., 2020), one may consult Ogata (2005) for more details. We have tested the accuracy of the method with the classical solution of Hantush (1964) in the revised manuscript.*

The authors simply point to Feng et al. (2020) and Liang et al. (2018) and do not discuss any details of the accuracy or convergence of their method for evaluating their "numerical" analytical solution (line 362). I would contend analytical solutions are more finicky and failure-prone than numerical models, so they require more careful scrutiny. The convergence of finite difference or finite element numerical models for solving confined groundwater flow (linear diffusion

equations for a homogeneous problem) is pretty well-known and is not going to surprise anyone. Numerical models can also consider: 1) finite wellbore radius, 2) heterogeneity, 3) variable pumping rates, 4) nonlinearities (e.g., the equation of state for water). I think the authors could cut down the section that discusses general "problems" with numerical models (lines 85-109) to a sentence. They could also cut down the section that talks about how general their analytical solution is. Both these un-needed sections could be replaced with discussion about the "numerical" details of evaluating their solution, which would actually be useful to someone who was going to try to implement this (the current general discussion about how much better analytical solutions is than numerical solutions in general is not useful).

*Reply: The details of the accuracy or convergence of our method for evaluating the semi-analytical solution involve the method of de Hoog algorithm (De Hoog et al., 1982) for Laplace transformation and the method of Ogata (2005) for Hankel transformation, the details have been discussed thoroughly in those two references. We have also addressed this issue in the revised manuscript.*

*In addition, some sections as suggested will be also deleted in the revised manuscript.*

3) The authors claim they have created a general and useful solution, but they also use an infinitessimal wellbore with no wellbore storage. Wellbore storage is very important, especially if you can have any range of aquifer properties in all three layers. All wells experience wellbore storage to some degree (unless it is a constant-head pumping test), the balance of the volume in the wellbore interval to the formation storage properties indicates whether or not it is significant. This solution will only be correct in the limiting case of small wellbore storage. The authors admit this (lines 636-645), but indicate that that will be coming in the next analytical solution.

*Reply: The available solutions have been shown that the effect of wellbore storage can only be found at early pumping time. We will study its influence on drawdown response in three-layer aquifer system in future.*

4) Variability in the pumping rate is a trivial difference between this solution and other solutions. Since the solutions is completely linear, Duhamel's theorem can be used to superimpose solutions that are pulses in time, or other combinations of steps on and off. The authors should provide some data or an example where this type of behavior (exponentially declining pumping rate) occurs. It is only included here because it is a simple case to consider in Laplace space, not because it is physically meaningful.

*Reply: The latest literature for an exponentially declining pumping rate test can be found in Chen et al. (2020). In their study, a variable-rate pumping test was performed in a borehole (YLW02) in Yanglinwei Town, city of Xiantao, situated in the Jianghan Plain, Hubei Province, Central China. The upper aquitard of clay and the lower aquitard of silty-clay are separated by the pumped aquifer, actual pumping rate can be expressed by an exponentially decay function.*

*Some data or an example with an exponentially declining pumping rate are implemented in the revised manuscript.*

On behalf of the authors
Sincerely Yours,
Hongbin Zhan

References cited

Chen, C., Wen, Z., Zhou, H., Jakada, H., 2020. New semi-analytical model for an exponentially decaying pumping rate with a finite-thickness skin in a leaky aquifer. J. Hydrol. Eng., 25(8): 04020037.

De Hoog, F.R., Knight, J.H., Stokes, A.N., 1982. An improved method for numerical inversion of Laplace transforms. SIAM J. Sci. Stat. Comput. 3 (3), 357–366.

Feng Q., Luo Y., Zhan H., 2020. Three-dimensional response to a partially penetration well pumping in a general anisotropic three-layer aquifer system, Journal of Hydrology, 2020, 585, 124850.

Hantush, M.S., 1964. Hydraulics of wells. Adv. Hydrosci. 1, 281–432.

Lenoach, B., Ramakrishnan, T.S., Thambynayagam, R.K.M., (2004) Transient flow of a compressible fluid in a connected layered permeable medium. Trans Porous Media 57:153–169

Liang, X., Zhan, H., Zhang, Y.K., 2018. Aquifer recharge using a vadose zone infiltration well. Water Resour. Res. 54, 8847–8863.

Ogata, H., 2005. A numerical integration formula based on the Bessel functions. Publications of the Research Institute for Mathematical Sciences, 41(4), 949–970.